# Brain states govern the spatio-temporal dynamics of resting-state functional connectivity

Felipe Aedo-Jury[1,2]*, Miriam Schwalm[1,3], Lara Hamzehpour[1], Albrecht Stroh[1,2]*

[1]Institute of Pathophysiology, University Medical Center Mainz, Mainz, Germany; [2]Leibniz Institute for Resilience Research, Mainz, Germany; [3]Department of Biological Engineering, Massachusetts Institute of Technology, Cambridge, United States

**Abstract** Previously, using simultaneous resting-state functional magnetic resonance imaging (fMRI) and photometry-based neuronal calcium recordings in the anesthetized rat, we identified blood oxygenation level-dependent (BOLD) responses directly related to slow calcium waves, revealing a cortex-wide and spatially organized correlate of locally recorded neuronal activity (Schwalm et al., 2017). Here, using the same techniques, we investigate two distinct cortical activity states: persistent activity, in which compartmentalized network dynamics were observed; and slow wave activity, dominated by a cortex-wide BOLD component, suggesting a strong functional coupling of inter-cortical activity. During slow wave activity, we find a correlation between the occurring slow wave events and the strength of functional connectivity between different cortical areas. These findings suggest that down-up transitions of neuronal excitability can drive cortex-wide functional connectivity. This study provides further evidence that changes in functional connectivity are dependent on the brain's current state, directly linked to the generation of slow waves.

*For correspondence:
felipe.aedo@lir-mainz.de (FA-J);
albrecht.stroh@unimedizin-mainz.de (AS)

**Competing interests:** The authors declare that no competing interests exist.

## Introduction

Spatio-temporal dynamics of cortical excitability are constantly impacted by fluctuations of internally generated activity, even in the absence of external inputs. The resting-state of the cortical functional architecture is characterized by canonical 'default-mode *networks*' (*Fox and Raichle, 2007*; *Mitra and Raichle, 2016*; *Raichle, 2015*) which have been identified in humans (*Raichle, 2011*), primates (*Chauvette et al., 2011*) and rodents (*Gozzi and Schwarz, 2016*; *Jonckers et al., 2011*). However, the spatio-temporal features of these networks are not static (*Deco et al., 2013*). Alternating states of excitability, manifesting as different activity states of the brain, can dynamically occur, even in different regions at the same time, as seen in the awake condition (*Guo et al., 2004*). Different states of brain activation can influence responses upon incoming sensory afferents (*Pachitariu et al., 2015*; *Schwalm et al., 2017*) and are accompanied by alterations in the spatiotemporal pattern of neuronal activity in many brain areas (*Pais-Roldán et al., 2020*). These alterations can be observed in local electrophysiological (*Constantinople and Bruno, 2011*; *Deco et al., 2013*; *Gozzi and Schwarz, 2016*) and optical (*Jonckers et al., 2011*; *Matsui et al., 2016*) recordings, as well as in large-scale readouts of brain activity, such as BOLD fMRI (*Kalthoff et al., 2013*; *Lewis et al., 2009*; *Staresina et al., 2013*). The ubiquity of fluctuating activity states which influence ongoing brain dynamics calls for a state-dependent assessment of resting state functional connectivity (*Tagliazucchi and Laufs, 2014*).

Recently, using local and brain-wide readouts, we have identified two activity states which were shown to occur under different types of anesthesia used for small animal functional magnetic

resonance imaging (fMRI) studies (*Schwalm et al., 2017*). In a state of persistent activity, which can occur during periods of light anesthesia, sedation or during awake periods (*Constantinople and Bruno, 2011*), neurons are rather depolarized, sparsely active, leading to temporally dynamic, modality specific, network configurations (*Barth and Poulet, 2012*; *D'Souza et al., 2014*). In contrast to persistent activity, stands the bimodal activity pattern of slow oscillations, or slow waves which have been extensively described under various conditions (*Chauvette et al., 2011*; *Ma and Zhang, 2018*; *Petersen et al., 2003*; *Sanchez-Vives et al., 2017*; *Sanchez-Vives and McCormick, 2000*; *Schwalm et al., 2017*; *Seamari et al., 2007*; *Stroh et al., 2013*; *Zucca et al., 2017*), but only most recently in the framework of BOLD fMRI (*Chang et al., 2016*; *Liu et al., 2011*; *Mitra et al., 2015*; *Schwalm et al., 2017*). Their corresponding low-frequency component ranges at 0.2–1 Hz, reflecting alternating activity patterns: active ('*up*') periods in which cells are depolarized and fire action potentials in temporally restricted periods, and silence ('*down*') periods with rather hyperpolarized membrane potentials and an almost complete absence of neuronal activity (*Destexhe et al., 2007*; *Steriade et al., 2001*). Slow waves can occur rather locally (*Nir et al., 2011*) or can spread over the entire cortex (*Ma and Zhang, 2018*; *Schwalm et al., 2017*), eventually allowing activity to propagate (*Sanchez-Vives et al., 2017*). In our previous work, we found that independent component analysis (ICA) for data obtained during persistent activity revealed canonical resting-state associated networks, such as the default mode (*Lu et al., 2012*; *Schwalm et al., 2017*) or different sensory networks (*Liang et al., 2013*; *Pawela et al., 2008*). Defining different states of brain activity by their local and global features of network dynamics by no means entails that their behavioral correlates, as for example in an anesthetized condition versus natural sleep, would be identical. However, defining features of slow wave activity, such as its bimodality, are resembled in a plethora of different conditions (*Bullmore and Sporns, 2009*; *Ma and Zhang, 2018*; *Mitra et al., 2018*; *Stroh et al., 2013*), likely because the fundamental difference between fast and slow timescale cortical dynamics is regulated by distinct mechanisms at the single neuron level (*Okun et al., 2019*).

Functional connectivity analysis for resting-state fMRI is used to identify and characterize neural networks in health and disease (*Buckner et al., 2005*; *Woodward et al., 2011*), during learning (*Lewis et al., 2009*) or memory consolidation (*Staresina et al., 2013*). The possibility to combine resting-state fMRI with invasive recordings of brain activity constitutes a pivotal branch of translational research (2018). Despite evidence of subtle changes in anesthesia level being able to dramatically change brain connectivity (*Grandjean et al., 2014*; *Hutchison et al., 2014*; *Zhang et al., 2019*), little has been investigated on how the brain's current state impacts functional connectivity derived from blood oxygenation level-dependent (BOLD) activity. While variations in functional connectivity have been related to cognitive state in humans (*Finn et al., 2017*), the effect of neurophysiologically defined states on functional connectivity in the cortex remains unknown. Directly relating functional connectivity to defined brain states may have explanatory value for a broad range of functional brain connectivity studies. We propose that connectivity measures in different conditions such as different anesthetics (*Bukhari et al., 2017*; *Kalthoff et al., 2013*; *Liu et al., 2013*; *Nasrallah et al., 2014*; *Paasonen et al., 2018*; *Shin et al., 2016*) can be classified based on the respective activity state of the brain. Based on these classifications, meaningful comparisons can be drawn both between functional connectivity signature in rodents as well as in humans.

Here, we find distinct BOLD functional connectivity patterns for a state of persistent activity and for a state dominated by the occurrence of slow waves. We identified BOLD correlates revealing compartmentalized network activity during the persistent activity state and a cortex-wide component during slow wave activity. Furthermore, we show a cortex-wide functional coupling of brain activity during slow wave activity which has its origin in transitions from population *down* to population *up* states which occur during ongoing slow wave activity. Our data reveals the imminent consequences of each slow wave event for the configuration of dynamic functional networks during this type of brain-wide activity.

## Results

We conducted ICA of rat resting-state fMRI signals during isoflurane-induced slow wave activity and in a medetomidine-induced persistent activity. For slow wave activity data, we found a characteristic

cortex-wide component as characterized previously (*Schwalm et al., 2017*). Conversely, during persistent activity, this cortex-wide component was absent and canonical independent components of well-compartmentalized default mode networks were prevailing. While we found a cortex-wide component in all animals during isoflurane-induced slow wave activity (*Figure 1A*), the identified components for persistent activity resembled those described by others for awake (*Becerra et al., 2011*; *Paasonen et al., 2018*) or sedated rodents (*Bukhari et al., 2018*; *Schwalm et al., 2017*). We identified default mode network activation in 14 of 15 animals (*Figure 1B*), a bilateral auditory component in 11 of 15 medetomidine-sedated animals (*Figure 1C*) and a bilateral component of the insula in 9 of 15 animals (*Figure 1D*). All the selected components of each animal can be observed in *Figure 1—figure supplement 1*. Similarly, as it has been reported in previous work (*Hsu et al., 2016*), we identified four default mode network regions: left and right posterior parietal cortex, anterior and posterior insular cortex and orbital/prelimbic cortex. To corroborate the difference in the network organization between both brain states, we then manually defined nine anatomical regions of interests (ROIs) based on the z-scores maps of the persistent activity components. We run a functional connectivity analysis of each ROI for each individual and plotted the resulting matrix. When both matrices are compared, it can be observed that there is a wide distribution of significant connectivity among the different networks (delineated by the black squares) during slow wave activity (*Figure 1E*) compared with the high specificity showed during persistent activity (*Figure 1E*). These results indicate that there are marked differences in the network configurations between both types of activity.

## Persistent and slow wave activity display different functional connectivity networks

To assess whether the cortical connectivity patterns previously described belong to those particular networks or are illustrating a general cortical phenomenon, we obtained the functional connectivity signature of the cortical networks during slow wave and persistent activity. We parcellated the brain based on anatomical landmarks using the Valdes-Hernandez et al template for rats (*Valdés-Hernández et al., 2011*) and correlated the BOLD signal of 96 cortical structures using the average signal from the white matter structures, ventricles and the animal's breathing rate as covariables in both conditions (*Figure 2A*). The name and order of cortical areas can be found in *Figure 2—figure supplement 1*. We plotted the cumulative distribution function (CDF) for the r-values of each matrix (n = 15) to establish whether a difference in the number and strength of significant correlations can be found. The cutoff was calculated for each matrix using FDR (mean 0.379 ± 0.041 s.e.m.). The CDF values differed significantly between both activity states (paired t-test, t(14) = 6.01 p<0.001) at the mean of the cutoff (*Figure 2B*). The number of correlated pairs for each individual at its particular cutoff was significant (paired t-test, t(14) = 6.502, p<0.001) (*Figure 2C*). These results suggest that slow wave and persistent activity are related to different functional connectivity patterns. To illustrate this point, we plotted connectivity patterns between regions in both brain states (*Figure 2C and D*), from this diagram it is possible to infer that during persistent activity, functional connectivity is more compartmentalized than during slow wave activity.

Population activity during the presence of slow waves is characterized by long periods of quiescence followed by short lapses of neural activity corresponding to a propagating wave (*Stroh et al., 2013*). Frequency and power distributions of the BOLD signal and their relationships to functional connectivity have been studied before with respect to specific anesthetic conditions (*Grandjean et al., 2014*; *Paasonen et al., 2018*). fALFF analysis allows to determine the fractional power of the BOLD signal for a particular region of interest (ROI) (*Zou et al., 2008*). To rule out that the connectivity values observed during slow wave activity are due to the spurious consequence of an increase in power, we performed a fALFF analysis to investigate the differences in the BOLD power (*Figure 3A*), showing that during slow wave activity there is a significant shift to lower power values in the distribution of the fALFF curves which were created with the 96 cortical ROIs of each animal (paired t-test, t(14) = 3.11, p=0.008). To test the dependence of the strength in connectivity from the distance of cortical areas, we correlated the Euclidean distance between pairs with the significant r-scores of the correlation of those pairs (*Figure 3B*). The closer to 90° angle the slope of those correlations gets, the more dependent of the Euclidean distance their correlation becomes. We found, that this is indeed the case for slow wave activity recordings (*Figure 3C*), whose slopes are significantly lower than those of data recorded during persistent activity (paired

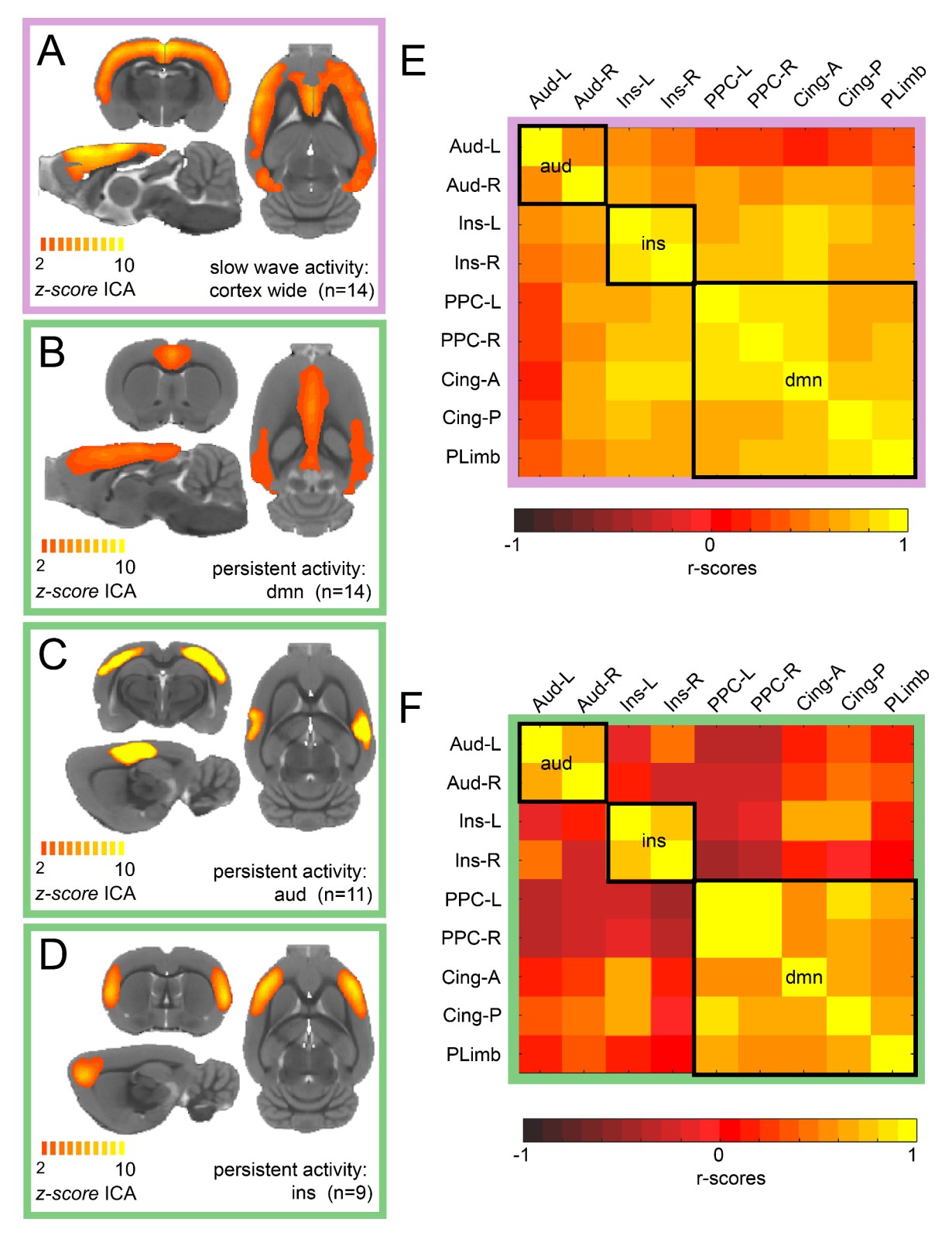

**Figure 1.** ICA reveals distinct components for slow wave and persistent activity. (**A**) Average z-score maps of the cortex-wide component during slow wave activity. B-D. Average z-score maps of two components found in 14, 11 and 9 animals respectively (**B** Default mode network, **C** auditory component and **D** insula activation) during persistent activity. A detailed diagram with all the z-scores maps subject by subject can be found in *Figure 1—figure supplement 1* (**E**) Resting-state functional connectivity matrix during slow wave activity from the nine regions of interest (ROIs) found

*Figure 1 continued on next page*

Figure 1 continued

in the ICA of persistent activity (**B–D**). Black squares outline the three networks. The ROIs are auditory cortex left and right (Aud-L, Aud-R), Insula left and right (Ins-L, Ins-R), posterior parietal cortex left and right (PPC-L and PPC-R), andterior and posterior cingulate cortex (Cing-A and Cing-P) and orbital/prelimbic cortex (PLimb). F. Same analysis as in E but for the signal obtained during persistent activity.

The online version of this article includes the following figure supplement(s) for figure 1:

**Figure supplement 1.** Individual z-scores obtained from the ICA for each animal shown in transversal view of an atlas template.

t-test, $t(14) = 5.43$, $p<0.001$). These results corroborate that the BOLD signal during slow wave activity is governed by different dynamics than the BOLD signal during persistent activity. Furthermore, during slow wave activity, BOLD-driven connectivity seems to be less compartmentalized and shows a stronger dependency on the distance between areas than it is the case for persistent activity.

## Cortical small world network architecture characterizes persistent activity and disappears during slow wave activity evidenced by graph theory analysis

To assess the organizational nature of the cortical networks identified by our previous analyses, we applied graph theory methods (*Rubinov and Sporns, 2010*). We first computed the modularity degree of the cortical networks identified during persistent and slow wave activity respectively. Modularity is a quality index for a partition of a network into non-overlapping communities (*Schwarz et al., 2008*). Therefore, in a brain with more compartmentalized networks this measure is higher than in one that shows a less selective network configuration. When we compared modularity values from networks identified for slow wave activity to those of networks identified for persistent activity, we indeed observed significantly higher values for persistent activity (paired t-test, $t(14) = 4.01$, $p=0.001$; *Figure 4A*), resembling results previously reported by others (*D'Souza et al., 2014*). Global and local efficiency are two important characteristics to differentiate small world from random networks. Small world networks exhibit high values for both measurements, whilst randomized networks show high values in global but low values for local efficiency (*Latora and Marchiori, 2001*). When we compared these measures, we did not find significant differences in the global efficiency scores (*Figure 4B*; paired t-test, $t(14) = 0.389$, $p=0.703$), but we found significantly lower scores for slow wave activity than for persistent activity in the local efficiency measurement (*Figure 4C*; paired t-test, $t(14) = 5.869$, $p<0.001$). It is well documented that both, small world and random networks show a short characteristic path length, whilst only small world networks show a high clustering coefficient (*Watts and Strogatz, 1998*). In our analyses, we found a significantly lower clustering coefficient for networks dominated by slow wave activity (*Figure 4D*; paired t-test, $t(14) = 3.789$, $p=0.002$), but no significant differences between networks in slow wave and persistent activity for characteristic path length (*Figure 4E*; paired t-test, $t(14) = 0.636$, $p=0.535$). This suggests that during slow wave activity the cortex rather behaves like a random network, whilst during persistent activity it exhibits small world network features (*Figure 4F*). In summary, graph theory analysis indicated a different functional connectivity pattern for persistent and slow wave activity. For persistent activity canonical well compartmentalized, small world networks seem to play a dominating role, while during slow wave activity a highly interconnected randomized network is prevalent.

## Different activity states of the brain are reflected by changes in functional connectivity

Functional connectivity patterns in the rodent brain can change dramatically depending on the anesthetics used (*Bukhari et al., 2017*; *Matsui et al., 2016*; *Wu et al., 2017*). In order to demonstrate our effects to be independent of the anesthetic agent, we compared the medetomine-induced sedation persistent activity to a condition of low isoflurane anesthesia (0.5% - 0.7%) that prevents animals of entering slow wave activity (n = 6 animals), hypothesizing that both protocols lead to a similar functional state, that is persistent activity. In the fiber photometry recordings (*Figure 5A*), it becomes apparent that the population dynamics recorded under this low isoflurane anesthesia regime resemble the ones obtained under medetomidine, rather than showing similarity to the ones recorded under higher isoflurane concentrations (*Figure 5B*). To quantify whether this observation is

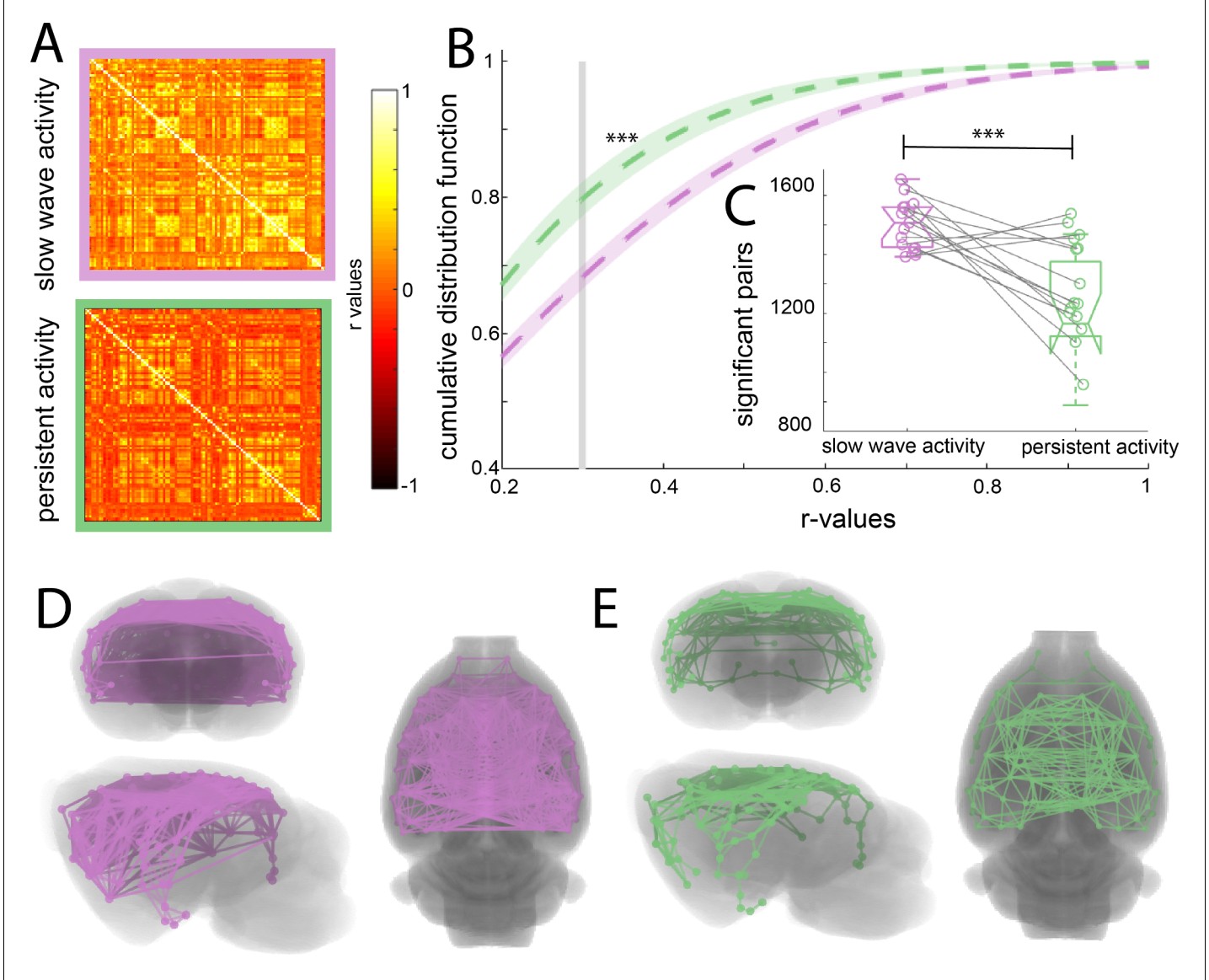

**Figure 2.** Increased functional connectivity during isoflurane-induced slow wave activity compared to persistent activity related to medetomidine sedation. (A) Average of the correlation matrices (n = 8) of the mean BOLD signal in 96 cortical regions for slow wave activity (magenta) and persistent activity (green). (B) Cumulative distribution curve for the r-values of the correlations. The vertical gray line lies at the mean of the FDR values used as a cutoff to identify significant correlations, ***p<0.001, paired t-test. (C) Box plot of the amount of significant correlations for each individual animal in both activity states ***p<0.001, paired t-test, vertical dashed line represents the data distribution, central horizontal bar the median and the two extremes horizontal bars point the interquartile range. (D) Diagram of the significant cortical connections during slow wave activity. Each line represents significant connections between cortical nodes (96 in total). Circles signaling the nodes are plotted in the center of the ROI. (E) Similar than D but for the persistent activity networks.

The online version of this article includes the following figure supplement(s) for figure 2:

**Figure supplement 1.** Cortical areas used for connectivity analysis.

meaningful for the BOLD signal, we computed the functional connectivity analysis in the same way as described for the previous experiments and obtained the connectivity matrices (*Figure 5C*). We computed matrix similarity analysis for each of the six matrices against the mean matrix for slow wave and persistent activity to determine if the connectivity patterns obtained under the low isoflurane concentration rather resemble the ones previously measured under medetomidine, (persistent activity) or the ones measured under higher isoflurane concentrations (slow wave activity). Similarity values can be compared extracting the eigen values of the matrix and then using *Frobenius norm* in

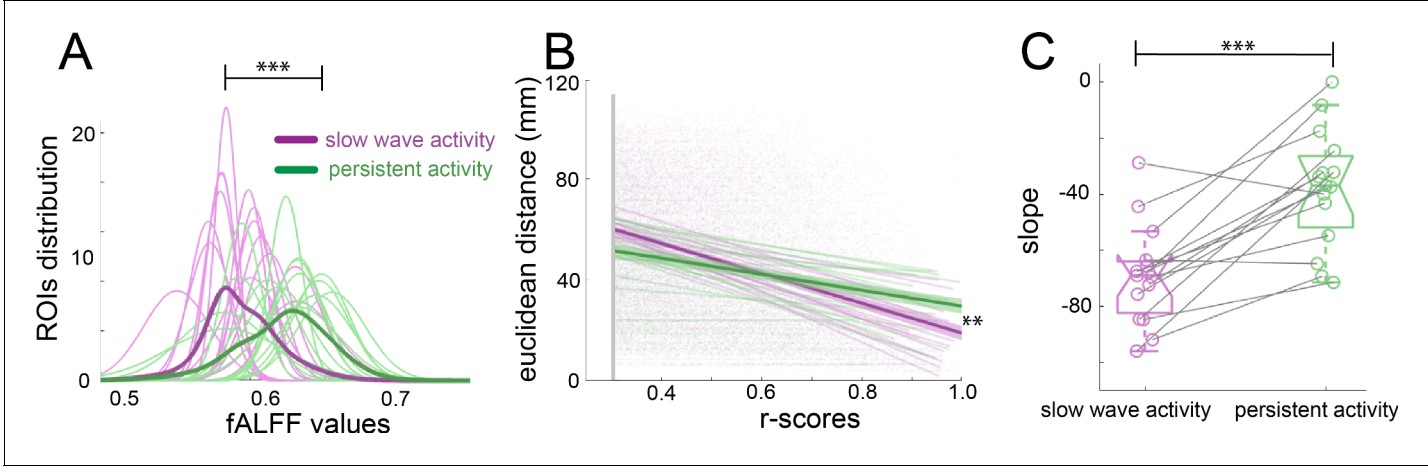

**Figure 3.** Network dynamics differ for slow wave and persistent activity. (A) Plot of the fALFF analysis. The fALFF value was obtained for each region of interest and then a normal distribution was fit with the values of the 96 cortical ROIs for each individual during slow wave (n = 15, magenta) and persistent activity (n = 15, green), ***p<=0.001. The average values of the distribution for each condition is represented by the thick line. (B) Correlation between the r-scores of each pair of cortical ROIs and their Euclidean distance, **p<=0.01. Each point corresponds to an r-score of a particular pair of ROIs. Linear correlation between ROIs and distance is plotted for persistent (n = 15, green) and slow wave activity (n = 15, magenta). The means of the correlations for each condition and the standard error of the mean are plotted in thicker lines. (C) Box plot of the slopes for each individual plotted in B. ***p<=0.001, paired t-test (n = 15). vertical dashed line represents the data distribution, central horizontal bar the median and the two extremes horizontal bars point the interquartile range.

order to calculate their similarity. The lower the obtained value, the greater is the similarity between two matrices. In this case, the resulting values related to persistent activity induced by low isoflurane, were significantly lower when compared with the ones for persistent activity induced by medetomidine sedation than when compared to the ones related to slow wave activity under high isoflurane concentration (*Figure 5D*; paired t-test, t(5) = 5.19, p=0.003). We used the same matrix similarity computation in a dynamic connectivity analysis. Therefore, we compared the initial 5 min of recordings with a moving average of 5 min in steps of 1 min. The results show that in the case of persistent activity induced by medetomidine and by low isoflurane concentrations, network configurations were significantly more stable than in the case of slow wave activity induced by high isoflurane concentrations (*Figure 5E*). This indicates that the network dynamics during slow wave activity might be related to a constant reorganization of connectivity patterns, probably due to the frequent changes in the rate of population *down-up transitions*.

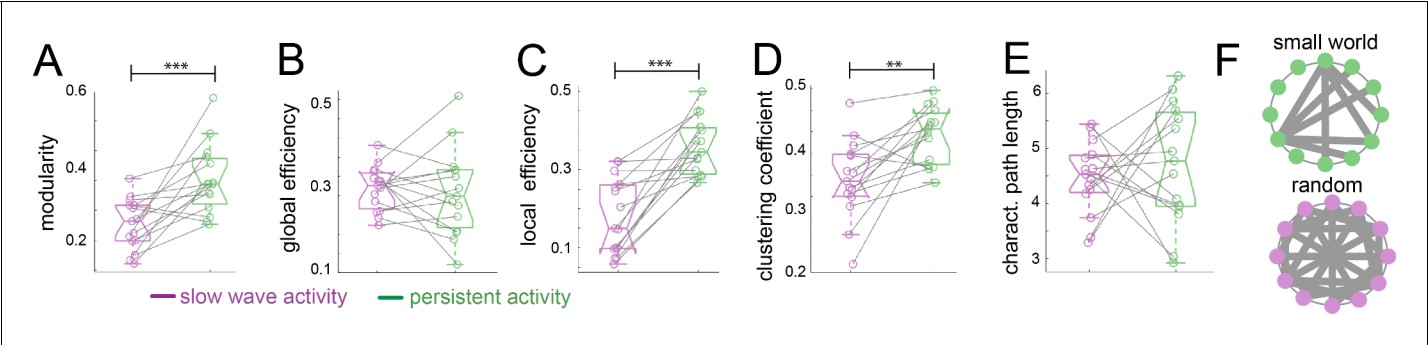

**Figure 4.** Graph theory shows a random network signature during slow wave activity. Legend: *=p < 0.05; **=p < 0.01; ***=p < 0.001 (paired t-test n = 15); slow wave activity (magenta), persistent activity (green). vertical dashed line represents the data distribution, central horizontal bar the median and the two extremes horizontal bars point the interquartile range. (A) Modularity values. (B) Global efficiency values. (C) Local efficiency values. (D) Clustering coefficients. (E) Characteristic path lengths. (F) Diagram representing a characteristic configuration of small world (green) and randomized networks (magenta).

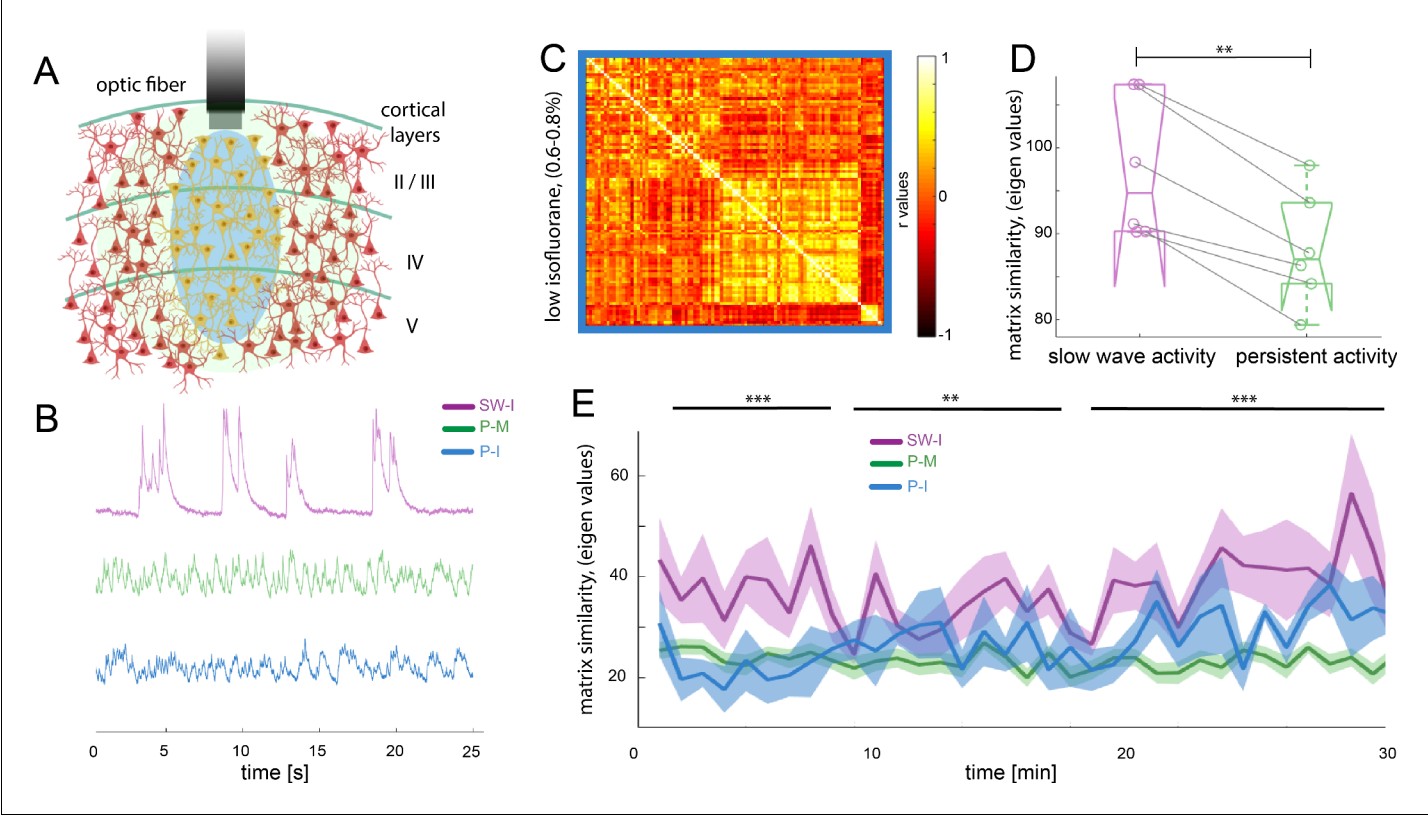

**Figure 5.** Different activity states of the brain result in characteristic functional connectivity matrices. (**A**) Calcium fiber photometry schematic. OGB-1 was bolus-injected and an optic fiber with a diameter of 200 µm was implanted at a cortical depth of about 300 µm. Blue light was used for excitation of OGB-1. Emitted fluorescence, comprising the changes in cytosolic calcium concentration of the local neural population was collected by the same fiber, and recorded by the fiber optometer. (**B**) Characteristic signal traces of photometry recordings during slow wave activity induced by high isoflurane (SW-I, magenta), and persistent activity induced either by medetomidine (P-M, green) or by lower concentrations of isoflurane (blue, P–I). (**C**) Average of the correlation matrices (n = 6) of the mean BOLD signal in 96 cortical regions under low isoflurane concentration. (**D**) Matrix similarity analysis of the low isoflurane (P-I n = 6) persistent activity experiments compared with the average matrix generated under high isoflurane related slow wave activity (SW-activity, magenta) and the medetomidine-induced persistent activity (P-M, green), **p<0.01, (n = 6, paired t-test). (**E**) Matrix similarity observed under the three conditions under a dynamic connectivity analysis. Lines represent the average of individual variability for slow wave activity (SW-I, n = 15, magenta), persistent activity induced with medetomidine (P-M, n = 15, green) and persistent activity induced with isoflurane (P-I, n = 6 blue), the semitransparent stripe correspond to the standard error of the mean for each condition. The matrices were generated for a 5-min period with steps of 1 min and were compared with the average matrix generated in the first 5 min for each individual animal, **p<0.01 (paired t-test n = 15 in SW-I and P-M conditions).

## Population down-up transitions drive functional connectivity during slow wave activity

Slow wave activity has been proposed to constitute the default activity pattern of the cortex, which is present during deep slow wave sleep or when connections with the thalamus are physically removed (*Sanchez-Vives et al., 2017*). Local occurrences of slow wave activity have also been shown to take place in cortical circuits during awake periods (*Vyazovskiy et al., 2011*). This bimodal pattern – periods of neural quiescence interrupted by short bursts of activity - generates waves of activity that can propagate across the cortex (*Busche et al., 2015*; *Stroh et al., 2013*). We recently showed that these waves generate a cortex-wide increase in BOLD activity (*Schwalm et al., 2017*) which may also be responsible for a cortex-wide increase of functional connectivity during slow wave activity. To test this, we analyzed the previously obtained functional connectivity results during slow wave activity while using the cortex-wide component as a covariable, since this component represents a direct correlate of local slow waves (*Schwalm et al., 2017*). If the high connectivity signature found during slow wave activity is related to population *down-up* transitions, the cortex-wide component as a covariable should remove any correlation that occurs due to such *down-up* transitions

observable in the population signal. Following this rationale, we expected that once the *down-up* transition covariable is removed from the connectivity analysis, correlation values should significantly drop (*Figure 6Ai*). Therefore, the cumulative distribution function of the r-values obtained using the cortex-wide component as a covariable (CWCcov) should be shifted to the left compared with those obtained using the original cortex inverted signal of the cortex-wide component (CWCinv). The CDF data (same analysis as in *Figure 2B*) showed significantly higher values for the CWCcov condition at the mean of the cutoff (*Figure 6Aii*) (paired t-test, t(14) = 4.96, p<0.001). Furthermore, the amount of significantly correlated pairs for each individual at its particular cutoff was also significantly less in the CWCcov condition. (paired t-test, t(14) = 5.435, p<0.001).This suggests a relationship between slow wave activity-related *down-up* transitions in the population signal and the strength of cortical functional connectivity. Additionally, we co-registered calcium photometry (*Adelsberger et al., 2014*; *Grienberger et al., 2012*) and fMRI BOLD signals during slow wave activity in five animals (*Figure 6Bi*). We quantified the number of population *down-up* transitions in a period of 3 min in the fluorescence signal emitted by a neural population in visual cortex (V1) (*Figure 6Bii*, upper panel) and correlated it with the number of significant pairs from the functional connectivity analysis of the BOLD signal for the same time period (*Figure 6Bi*, lower panel). We quantified these values for 30 min-long recordings using samples of 3 min and a gap of 1 min in between them. The correlation between the amount of *down-up* transitions in the fluorescence signal and the BOLD pairs was significant for five of the six animals (p<0.001) (*Figure 6Bii*). In order to overcome a potential source of a spurious correlation, we run a permutation test on each resulting correlation. The permutation test allows to establish the baseline correlation between points and to calculate the distance of the obtained results from that correlation. In the five animals, the permutation test shows that the correlation r-scores are in the 5% of significant values (*Figure 6Bii*, normal distribution plots). Finally, we calculated the statistical value for the whole dataset using the recently described repeated measures correlation (*Bakdash and Marusich, 2017*) a statistical technique for determining the inter-individuals correlation for intra-subject multiple measurements (r(196) = 0.534, p<$10e^{-15}$). These results strongly suggest that slow wave activity-related *down-up* transitions are directly linked to the increase in cortical functional connectivity during the prevalence of slow waves.

## Dual-cortical calcium photometry during slow wave activity shows increased signal correlation in down-up transitions

While the previous results demonstrated the role of *down-up* transitions for functional connectivity during slow wave activity, they are based on hemodynamic signals rather than on a direct measure of neural responses. The occurrence of slow waves in a spatially confined neural population can be detected using fiber photometry measuring fluorescent signals emitted by cells previously stained with a calcium indicator (*Adelsberger et al., 2014*; *Grienberger et al., 2012*). To corroborate our results on the neural population level, we conducted experiments employing simultaneous optic-fiber-based calcium recordings in somatosensory (S1) and visual (V1) cortex (*Figure 7A,B*). Slow wave related *down-up* transitions can be easily identified and quantified (*Seamari et al., 2007*) and have a duration of 2 s on average (*Figure 7C*). The amplitude of the cross-correlation between these two signals is significant (*Figure 7D*), with a delay matching the propagation time of the wave propagating from V1 to S1 ((n = 6) 51.2 mm/s ± 8.1 s.e.m). To probe if *down-up* transitions are underlying the increase of functional cortical connectivity in our slow wave activity data, we filtered the fluorescence and the BOLD signal at different frequency intervals before running the cross-correlation analysis (*Figure 7B,D*). Filtering the signal induced a partial abolishment of the slow waves in particular bandwidths (*Figure 7C*) without interfering with the correlation of both cortical signals in other frequencies. Therefore, one should expect a decrease in the signal correlation on those frequencies when the slow wave activity is abolished. Results of six recordings in three animals showed that the cross-correlation peaks significantly decrease in the frequency interval for which the *down-up* transitions were filtered out (*Figure 7B,E*). The cross-correlation values were significantly lower when data was filtered between 0.01 and 0.4 Hz (paired t-test: t(5) = 4.26, p=0.008), 0.2–0.6 Hz (t(5) = 7.99, p<0.001), and 0.3–0.7 (t(5) = 7.28, p<0.001). These results support the ones obtained previously with BOLD connectivity analyses at the cortical population level, demonstrating *down-up* transitions to be the main contributor to an increase of cortical functional connectivity during slow wave activity.

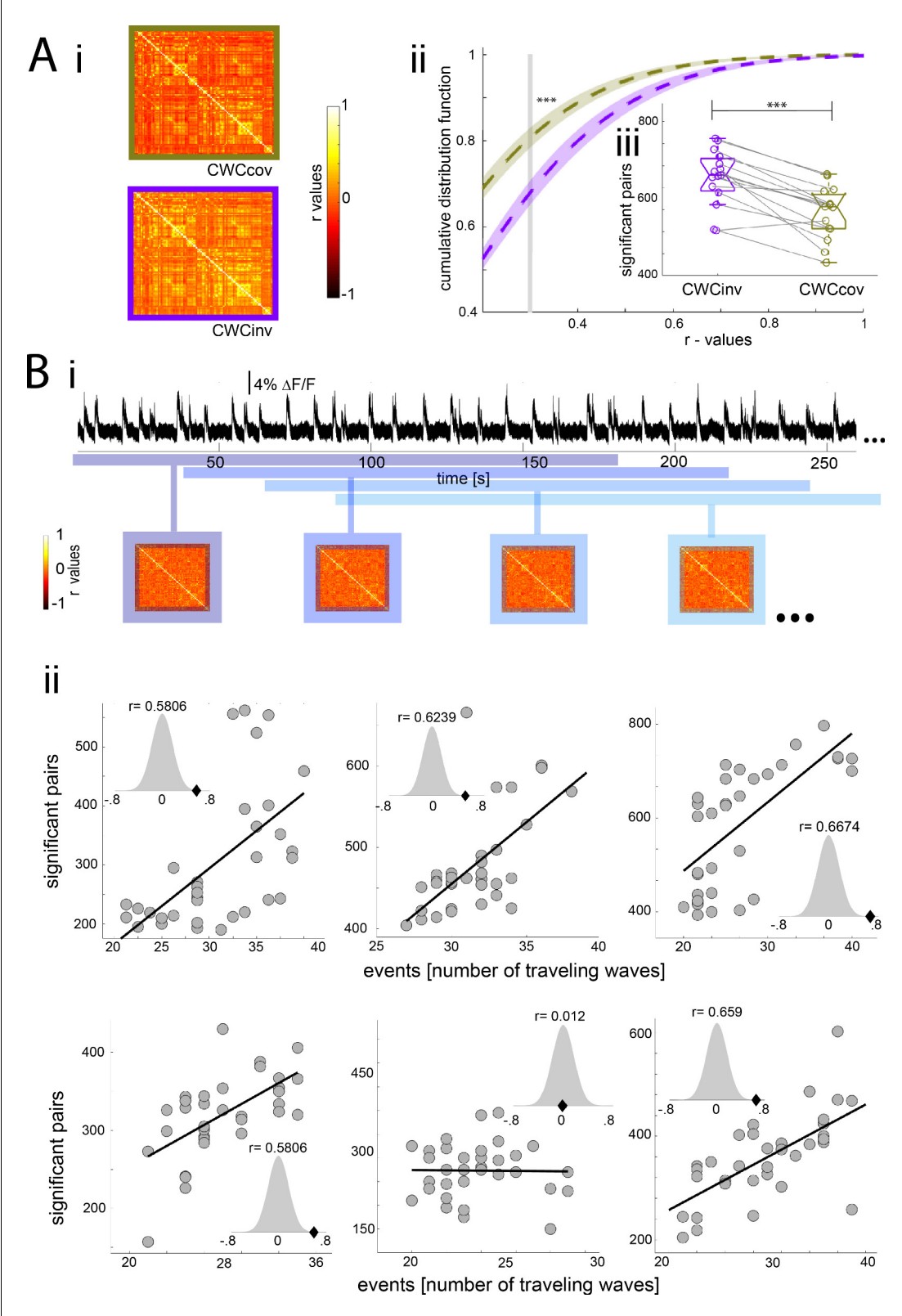

**Figure 6.** Population *down-up* transitions drive functional connectivity during slow wave activity. (**A**) Functional connectivity of the BOLD signal using the cortex wide component of the ICA as covariable (CWCcov): i- Average of the correlation matrices (n = 8) of the mean BOLD signal in 96 cortical regions during slow wave activity using the pan-cortical component of the ICA as covariable (CWCcov, purple frame) or the flipped values of the same component as control (CWCinv, brown frame). ii- Cumulative distribution curve for the r-values of the correlations during slow wave activity using the

*Figure 6 continued on next page*

*Figure 6 continued*

cortex-wide component of the ICA as regressor (CWCcov), or similarly using the flipped values of the same component (CWCinv). The vertical gray line lies at the mean of the FDR values used as cutoff to identify significant correlations, ***p<0.001 (paired t-test, n = 15). iii- Plot of the number of significant correlations for each individual in both conditions **p<0.01. (B) Correlation between the number of significant pairs in the functional connectivity matrix and the number of *down-up* transitions. i- Correlation scheme. The number of *down-up* transitions was quantified for the respective time period (upper panel) and then correlated with the number of significant pairs obtained from the functional connectivity matrix (lower panel). ii- 5 out of 6 animals analyzed showed a significant positive correlation between number of *down-up* transitions and number of significant functionally connected pairs. Histogram subplots indicate the results of the permutation test. This analysis corroborates the Pearson correlation showing the r-scores of the five significant cases to lie beyond 5% of significance in the normal distribution built with 10000 surrogates of shuffled data.

## Discussion

We demonstrated that different states of brain activity can modify properties of the functional connectome. ICA showed distinct functional networks for each of the investigated activity states. In the

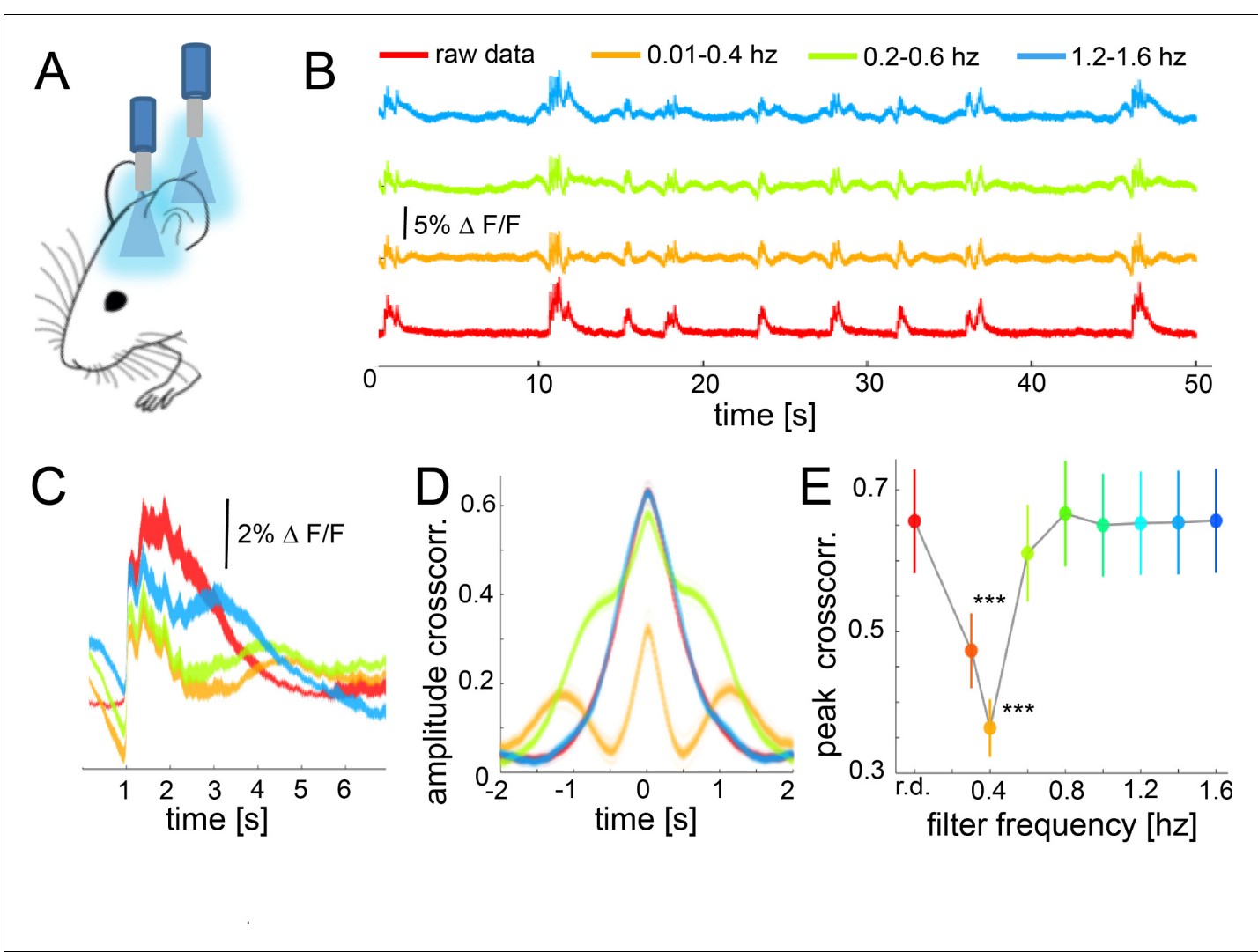

**Figure 7.** During slow wave activity, the correlation between distant optical recordings is driven by slow-wave-associated calcium waves. (A) Schematic of fiber photometry in cortical areas S1 and V1. (B) Exemplary fluorescent signal trace recorded from V1 filtered at different band-stop intervals, excluding those particular intervals from the recordings. (C) Mean of the *down-up* transitions for a trace (Δf/f) at different band stop filters. It can be observed how the amplitude, length and size of the slow waves are modified at each filtered bandwidth. (D) Cross-correlation results of a single recording filtered at different band-stop intervals (color code is the same than in subplot B). (E) Average of the cross-correlation peak values (n = 6) at different band-stop intervals, ***p<0.001 (Paired t-test, n = 6).

case of slow wave activity, a cortex-wide component was found in all animals, while for persistent activity, a clear compartmentalization of the cortical functional structures gave origin to well-determined functional networks, as the default mode or the auditory network. Notably, the relation of population activity to functional connectivity was not dependent on a specific anesthetic agent, suggesting that not the type of anesthesia used, but the induced activity state influenced resting-state networks. The functional connectivity analyses using cortical parcellation corroborated this concept, showing an increase in the connectivity r-scores as well as in the number of significant correlations during slow wave activity. Population activity dominated by slow waves is characterized by long periods of quiescence followed by short lapses of neural activity corresponding to a propagating wave (*Stroh et al., 2013*), as demonstrated by the simultaneous dual cortical calcium recordings. To rule out that the connectivity values observed during slow wave activity are due to the spurious consequence of an increase in power, we performed fALFF analysis and found significantly lower values for data obtained under slow wave activity, probably due to the long periods of quiescence observed in population slow wave activity.

## Spatiotemporal network dynamics

The high degree of compartmentalization in persistent activity dominated networks, observed in the ICA can be corroborated by calculating the dependency of the distance in the r-scores between two areas. In a highly compartmentalized connectome, this dependency should be lower than in a system which is highly interconnected (*Hagmann et al., 2008*). Indeed, when correlating the r-scores and the Euclidean distance of the network's pairs for persistent activity data, the correlation was significantly lower than for data obtained under slow wave activity. This finding suggests, that indeed, the functional architecture during persistent activity is dominated by specific compartmentalized network nodes, which are not necessarily spatially close, such as the default mode network. In sharp contrast, during slow wave activity, we observed a strong distance dependency, in line with a propagating wave of activity – a slow wave event – eventually recruiting large parts of the cortex. These propagating slow wave events might temporarily switch parts of the functional network from a hyperpolarized, rather quiescent down state into a depolarized *up* state of high excitability and thereby propagate over cortical areas (*Sanchez-Vives et al., 2017*; *Sanchez-Vives and McCormick, 2000*; *Steriade et al., 1991*).

Applying graph theory methods, we found that during persistent activity the functional network shows characteristic small world network dynamics, as they have been described previously for the sedated rat (*D'Souza et al., 2014*) and for the awake human brain (*Bullmore and Sporns, 2009*). During slow wave activity graph theory analysis demonstrated characteristics of highly interconnected random networks to be dominant, corroborating the idea of a lack of a spatially segregated functional connectivity signature in that state (*Okun et al., 2019*; *Spoormaker et al., 2010*).

One could argue that the difference in connectivity is reflecting a particular change in neural dynamics induced by a specific anesthetic regimen rather than presenting a correlate of a particular activity state, since it has been reported that functional connectivity patterns in the rodent brain change dramatically depending on the anesthetics used (*Bukhari et al., 2017*; *Matsui et al., 2016*; *Paasonen et al., 2018*; *Wu et al., 2017*). In order to rule out anesthesia-specific effects on connectivity, we demonstrated that decreasing the amount of isoflurane administrated to the animal led to a transition from slow wave to persistent activity. Indeed, population and functional connectivity dynamics observed under a low dose of isoflurane rather resembled the ones observed under medetomidine-induced persistent activity, than the ones observed during slow wave activity induced by high isoflurane anesthesia.

Recently, it is has been shown that spontaneous resting-state associated brain activity is non-random in the temporal domain and that this feature is well conserved for rodents and humans (*Ma and Zhang, 2018*). Interestingly, when we analyzed the temporal dynamics of connectivity patterns, we found that animals in persistent activity state, induced either by medetomidine or a low dose of isoflurane, showed rather temporally stable - persistent - connectivity dynamics (*Destexhe et al., 2007*; *Sanchez-Vives and McCormick, 2000*; *Steriade et al., 1991*). In contrast, animals in slow wave activity state showed fast, and temporally irregular transitions likely reflecting the bimodality of slow waves (*Stroh et al., 2013*). This suggests that individual specific events within slow wave activity could drive state-specific network signatures, while it has to be taken into account that the regularity of *down-up* transitions or slow wave events can depend on the overall

physiological condition of the animal, such as the depth of anesthesia (*Chauvette et al., 2011*). Although, there is abundant evidence on the relevance of, for example, infra-slow electrical brain activity for resting state connectivity (*Hiltunen et al., 2014*; *Krishnan et al., 2018*; *Mitra et al., 2018*), there so far has been no evidence linking different population activity states of the brain to changes in functional network configuration directly.

When we used the cortex-wide component as a covariable in the connectivity analyses, the connectivity values decreased significantly. This component represents a direct correlate of slow wave activity in a local neural population (*Schwalm et al., 2017*) and is therefore evident for the occurrence of population down-up transitions. A direct demonstration of the involvement of down-up transitions in the functional connectivity maps comes from the significant correlation between the degree of connectivity and the amount of transitions in 4 of 5 animals. To provide evidence for this hypothesis, we conducted simultaneous fiber photometry alongside fMRI measurements during slow wave activity, as previously implemented (*Schmid et al., 2016*). When the component of *down-up* transitions identified in the fluorescence signal was filtered out, we indeed found a decrease in correlation of cortical activity. These findings provide additional evidence that population *down-up* transitions can drive functional connectivity in the cortex.

It has been suggested that the generation and propagation of slow waves is depending on changes in cortical excitability during sleep (*Massimini et al., 2004*) or under anesthesia (*Ruiz-Mejias et al., 2011*), even differentially affecting cortical layers (*Capone et al., 2019*). Disrupted slow wave propagation can be reflective of pathophysiological conditions such as Alzheimer's disease (*Busche et al., 2019*). We therefore propose that distinct activity states of the brain, that is defined spatio-temporal dynamics of cortical excitability, drive the brain's functional connectivity. The here investigated rather discrete states – persistent versus slow wave activity, are characterized by particular and well-defined network signatures. The understanding of network dynamics underlying respective functional network patterns will reduce the variability found in resting-state data and increase the translational value of resting-state fMRI studies performed in rodents.

## Implications on resting state studies

Several studies have demonstrated that the functional state of the brain is dynamic over time, and that this is reflected in changes in functional connectivity (*Belloy et al., 2018*; *Grandjean et al., 2017*). These dynamic states are quantifiable, depend on ongoing activity and seem to show atypical dynamics in some neurological disorders (*Gutierrez-Barragan et al., 2019*). Degrees of wakefulness seem to play an important role in global functional connectivity (*Turchi et al., 2018*). One study showed that during resting-state recordings in humans, dynamic changes in functional connectivity occur depending on different wakefulness and sleep patterns (*Tagliazucchi and Laufs, 2014*). More recently, Laumann and colleagues showed that the dynamic variability over time in fMRI functional connectivity signatures can be explained by fluctuating sleep stages (*Laumann et al., 2017*).

Previous resting-state fMRI studies report that anesthetized animals show lower functional coupling than awake animals (*Vincent et al., 2007*; *Yin et al., 2019*). However, some studies show that default mode networks in rodents can be found even after the loss of consciousness (*Liang et al., 2012*; *Lu et al., 2012*), while others show a breakdown in the networks of resting state connectivity under anesthesia (*Boveroux et al., 2010*; *Stamatakis et al., 2010*). These apparent dissimilar results have been hypothesized to be dependent on the synchronization of infra-slow oscillatory activity and cross-frequency coupling (*Hutchison et al., 2013*). Our experiments demonstrate that the origin of these discrepancies may lie in the presence of different activity states of the brain. During persistent activity, compartmentalized patterns are detected, whilst during slow wave activity the degree of connectivity depends on the rate of slow waves generated in a particular period of time. Global signal removal has been shown to reduce connectivity for high isoflurane global synchrony conditions (*Grandjean et al., 2014*). This might indeed be because this technique is removing the underlying *down-up* transition related signals which are actually explaining a relevant part of the increased connectivity as our results show. Activity states of the brain as well as the rate of *down-up* transitions during slow wave activity in particular are dynamic and can be present under different experimental settings (*Vyazovskiy et al., 2011*). Here, we present data from strictly controlled conditions, in which the entire cortex seems to be engaged in either persistent or slow wave activity. However, in awake animals or during slow wave sleep, this might not be the case.

The presented results call for a brain state-informed analysis of functional connectivity in resting-state studies, both in humans and in animal models. Human subjects may transition between similar subtypes of these activity states when they transition between different levels of attention during an experiment or when they fall asleep (*Stevner et al., 2019*). Even more so, animals may react differently to the same anaesthetic regimen (*Vyazovskiy et al., 2011*) and therefore present a different underlying state of cortical activity while being subjected to the same anaesthesia condition. Similarly, it might be the case that at the beginning of an imaging experiment, an animal is in a state of persistent activity, and while the experiment progresses, shifts into a state of slow wave activity due to a change of underlying physiology or deepened anaesthesia over time. As our results show that this would lead to drastically different functional connectivity signatures, it is necessary to consider the particular activity state and global changes in cortical excitability before deriving conclusions about networks dynamics in any resting-state fMRI experiment.

# Materials and methods

## Key resources table

| Reagent type (species) or resource | Designation | Source or reference | Identifiers | Additional information |
|---|---|---|---|---|
| Genetic reagent (*Rattus norvegicus*) | Lewis rat | Janvier labs | LEW/OrlRj | Female 160–200 gr |
| Chemical compound, drug | PBS tabletts | gibco | 18912–014 | |
| Chemical compound, drug | xylocaine | AstraZeneca | PUN080440 | 2% |
| Chemical compound, drug | isoflurane | AbbVie | 8506 CHEBI:6015 | 1–1.55% |
| Chemical compound, drug | Oregon-Green BAPTA1 AM | Invitrogen Thermo Fisher scientific | O6807 | 1 mM |
| Chemical compound, drug | Glucose | B. Brown | 2355740 | 10% |
| Chemical compound, drug | Medetomidine hydrochloride | Dorbene vet. Zoetis | 08164–43 | 0.08 mg/kg/hr |
| Chemical compound, drug | Carbomer | Vidisic | 74013T296/52-DE | 2 mg/gr |
| Chemical compound, drug | NaCl | Fresenius Kabi France | 13KLP183 | 0.9% |
| Chemical compound, drug | Agarose | Sigma-Aldrich | CAS: 9012-36-6 | 2% |
| Software, algorithm | Brain Voyager 20.6 | Brain Innovation, Maastricht, Netherlands | | |
| Software, algorithm | Matlab R2018a | (The Mathworks, Inc, Natick, MA, USA) | | Codes available at https://github.com/Strohlab/connectivityelife copy archived at https://github.com/elifesciences-publications/connectivityelife |

## Animals

Experiments were performed on 25 adult female Lewis rats (>12 weeks old, 160–200 g), of which 15 were used for fMRI measurements, six were implanted with an optic fiber in the visual cortex for combined optic-fiber-based calcium recordings/fMRI measurements and four were used for optical recordings on the bench. Animals were housed under a 12 hr light–dark cycle and provided with food and water ad libitum. Animal husbandry and experimental manipulation were carried out according to animal welfare guidelines of the Johannes Gutenberg-University Mainz and were approved by the Landesuntersuchungsamt Rheinland-Pfalz, Koblenz, Germany (G14-1-040). fMRI data acquisition fMRI data acquisition for connectivity analyses was performed in two sessions for

each animal. For recording slow wave activity data sets, animals were anesthetized with isoflurane (1.2–1.8% in 80% air and 20% oxygen). For recording persistent activity data, animals were sedated with medetomidine (zoetis, bolus injection of 0.04 mg/kg i.p. followed by continuous i.p. infusion of 0.08 mg/kg/hr). In a second group (low isoflurane group), animals were sedated with isoflurane 0.5–0.7% in order to keep the animals in persistent activity. The order of the recording sessions was randomly assigned to each animal. Both sessions were separated by at least 1 week. Measurements were performed on a 9.4 T small animal imaging system with a 0.7 T/m gradient system (Biospec 94/20, Bruker Biospin GmbH, Ettlingen, Germany). Each experimental protocol consisted of T1-weighted anatomical imaging (image size 175 × 175×80, FoV 26 × 29×28 mm, resolution 0.15 × 0.165×0.35 mm), 30 min of resting state T2*-weighted images acquired with a single-shot gradient echo EPI sequence (TR = 1.5 s, TE = 14 ms and FA 65°, 320 × 290 $\mu m^2$ spatial resolution, slice thickness 0.8 mm, 34 continuous slices, 1200 acquisitions in eight animals and TR = 1 s in another seven animals) and a high-resolution in-plane T2-MSME images of the 34 slices (TE = 12 ms, 100 × 100 $\mu m^2$ spatial resolution, image size 256 × 256, slice thickness 0.8 mm, 34 continuous slices). The same protocol was used in the combined recordings with optic-fiber-based calcium recordings. Body temperature and breathing rate of the animal were monitored and recorded during the entire experimental procedure using an SA instruments small animal MRI-compatible monitoring system (Small Animal Instruments, Inc, Stony Brook, NY).

## fMRI analysis

### Data preprocessing

General Linear Model (GLM) fMRI data processing was performed using Brain Voyager 20.6 for windows (Brain Innovation, Maastricht, Netherlands). Preprocessing steps included slice scan time correction, 3D motion correction and smoothing of the images, using cubic-spline interpolation method and trilinear/sinc interpolation for the 3D motion correction. Datasets were smoothed using a 0.8 mm FWHM Gaussian Kernel. The initial five images were discarded to include only signals that have reached steady state. T1 images were manually aligned to the rat atlas MRI template of Valdes-Hernandez et al. (*Valdés-Hernández et al., 2011*). T2*-weighted images and high-resolution in-plane T2*-weighted images were then manually aligned to these T1 previously aligned to the MRI template. The same preprocessing steps were performed for the scans combined with fiber photometry recordings.

### Independent component analysis (ICA)

For ICA, the freely available software 'Group ICA of fMRI Toolbox (GIFT)', version 4.0b (http://mia-lab.mrn.org/software/gift/; RRID:SCR_001953) was used. Mask-calculation using the first image was employed and time-series were linearly detrended and converted to z-scores. For back-reconstruction, the 'GICA'-algorithm (*Calhoun et al., 2001*) has been used and all results have been scaled by their z-scores. We used the Infomax-algorithm with regular stability analysis without autofill. Images for the ICA were motion corrected and smoothed as described above for each animal. The number of ICs to calculate for each animal was set to 20 and 25 for checking whether some components were subdivided into partial networks. We did not find different components for the networks of interests between both analyses; thus, we used the 20 components. Breathing rate and the signal form ventricles were used a regressors to avoid spurious components. Components that showed clusters outside the brain or a distinctive single slide activity were considered as noise and discarded. For visualization purposes, the z-scores of each animal were averaged and then plotted over the template using the Brain Voyager software (*Figure 1*). The regions found in the persistent activity ICA were taken as ROI for a connectivity analysis. nine regions were identified base in the connectivity patterns. The time course from each of the nine preselected nodes was extracted, and computed the Pearson correlation between every pair of the nodes to form a correlation matrix. The average signal of the ventricles and white matter was extracted and together with the breathing signal added as a covariable in the correlation analysis. This step resulted in a 9 × 9 matrix for each rat, where r-scores represented the resting state functional connectivity strength between each pair.

## Connectivity analysis

Connectivity analyses were performed using custom Matlab scripts (The Mathworks, Inc, Natick, MA). After normalizing the BOLD-signal to its mean intensity, the cortical volume was parcellated in 96 regions of interest (ROIs) using the MRI template of *Valdés-Hernández et al., 2011*. The size of each region consisted in average on 2014 voxels. With an exponential decay distribution. 75% of the regions contained more than 300 voxels with a minimum of 102 voxels and a maximum of 10160. The mean signal for each area was calculated averaging the signal of each voxel. The average signal of the ventricles and white matter was extracted and together with the breathing signal added as a covariable in the correlations. In a second connectivity analysis (*Figure 6*), the time course of the cortex-wide component was added as a covariable, either in its original form or temporally reversed as a control. Then, partial correlation coefficients were calculated for each pair of ROIs. The significance cutoff for each pair of ROIs was calculated independently for each matrix (each individual) using false discovery rate (FDR) correction (mean of cuttof: $0.348 \pm 0.045$ s.e.m). The FDR was performed on each matrix using the function *mafdr* from matlab based on the method introduced by *Storey, 2002*. Thus, any value above the calculated FDR was considered as a significant connection between 2 ROIs.

## fALFF analysis

The amplitude of low-frequency fluctuations (ALFF) of the resting-state fMRI signal is defined as the power of the signal within a certain frequency range and measures the intensity of regional spontaneous brain activity in humans (*Zang et al., 2007*) as a measure of local metabolism in a particular region. To reduce sensitivity to physiological noise, *Zou et al., 2008* introduced fractional ALFF (fALFF), which is defined as the ratio of low-frequency amplitudes (0.01–0.08 Hz) to the amplitudes of the entire frequency range (0–0.25 Hz). Similarly, we conducted fALFF analysis using the ratio of 0.01–0.1 Hz for the entire frequency range 0–0.25 Hz. We calculated the fFALFF value for each voxel and then averaged those values for each ROI and compared the distribution of the 96 ROIs for both investigated activity states (slow wave activity and persistent activity).

## Euclidean distance

The correlation between r-scores and the Euclidean distances between the ROIs were calculated between each pair of ROIs taking the average coordinates of the voxels belonging to each ROI as its centroid.

## Graph theory analysis

Adjacency matrices were constructed starting from correlation matrices of the functional connectivity analysis, and after thresholding (threshold was given for each matrix based on the FDR correction previously calculated, see *connectivity analysis* above) were configured as *Weighted Undirected Networks* taking into account only the positively correlated values. This construction of matrix adjacency has been pointed as necessary when functional MRI datasets are analysed (*Rubinov and Sporns, 2010*). The adjacency matrices construction and the network measures calculated were all derived from the adaptation of the Complex Networks Analysis toolbox *Functional Brain Connectivity* for Matlab (*Rubinov and Sporns, 2010*). Network *nodes* in our case were ROIs, and networks *links* were the magnitudes of temporal correlations between ROIs, obtained by correlation coefficients calculated previously.

## Matrix similarity analysis

Matrix similarity was calculated using the Frobenius Norm of the eigenvalues of each matrix. This calculation gives a measure of the distance between the two vectors resulting from the eigenvalues of the matrix. The closer to zero the obtained norm is, the closer are the vectors and therefore, the more similar are the matrices.

## Calcium indicator injections and optic fiber placement

For surgical procedures including staining with fluorescent calcium indicator and optic fiber implantation, rats were anesthetized with isoflurane (Forene, Abbvie, Ludwigshafen, Germany), placed on a warming pad (37° C), and fixed in a stereotactic frame with ear and bite bars.

The skull was exposed, dried from blood and fluids, and leveled for precise stereotactic injections. Under a dissection microscope a small craniotomy was performed with a dental drill (Ultimate XL-F, NSK, Trier Germany, and VS1/4HP/005, Meisinger, Neuss, Germany). The fluorescent calcium sensitive dye Oregon Green 488 BAPTA-1AM (OGB-1, Invitrogen, Life Technologies, Carlsbad, CA) was prepared as described previously (*Garaschuk et al., 2006*), filtered, and injected into primary visual cortex (V1; −5.5 mm AP, +3.8 mm ML, −0.5,–0.7 and 0.9 mm DV) for the experiments combining fMRI and calcium photometry, and a second craniotomy and injection were performed in the primary somatosensory cortex front limb area (S1FL; 0 mm AP, +3.5 mm ML, −0.5,–0.7 and 0.9 mm DV;) for the bench experiments including optical recordings in the two cortical areas. After injection of the indicator, the pipette was held in place for 2 min before slowly retracting it from the tissue. After removing the cladding from the tip, an optic fiber with a diameter of 200 µm was inserted perpendicular to the dura above the OGB-1 stained area, typically at 300 µm and fixed to the skull with UV glue (Polytec, PT GmbH, Waldbrunn, Germany). A detailed description of these procedures can be found in *Stroh et al., 2013* and *Schwalm et al., 2017*.

### Fiber photometry calcium recordings

For optic-fiber-based photometry calcium recordings, custom-built setups were used (FOM and FOMII, npi Electronic Instruments, Tamm, Germany), as reported previously (*Stroh, 2018*; *Schwalm et al., 2017*). The light for excitation of the calcium indicator OGB-1 was delivered by a 650 mW LED with a nominal peak wavelength of 470 nm. LED power was controlled by an adjustable current source. The light beam was focused by means of a fixed focus collimator into one end of a multimode optic fiber which was connected to the system by an SMA connector. The recorded fluorescence signals were digitized with a sampling frequency of 2 kHz using a multifunction I/O data acquisition interface (Power1401, Cambridge Electronic Design, Cambridge, UK) and its corresponding acquisition software (Spike2).

### Calcium recordings analysis

Data was read into Matlab (The Mathworks, Inc, Natick, MA, USA) and downsampled from 2 kHz to 1 kHz by averaging two adjacent values. The baseline of the fluorescence signal was determined and corrected by using the Matlab function *msbackadj* estimating the baseline within multiple shifted windows of 2500 datapoints width, regressing varying baseline values to the window's datapoints using a spline approximation, then adjusting the baseline in the peak range of the fluorescent signal. This method provides a correction for low-frequency drifts while avoiding contributions of signal-of-interest events. We employed a procedure based on exponential moving average (EMA) filters, originally established to identify slow waves in electrophysiological recordings (*Seamari et al., 2007*). Onsets of slow calcium waves were defined as signal timepoints exceeding the threshold (70%), termination of calcium slow waves was defined as signal timepoints dropping below 50% of the threshold value. Further, the detected activity was post-processed in the following order: 1) calcium waves separated by a time-interval below 500 ms were interpreted as one wave; 2) calcium waves with a duration of less than 300 ms were discarded; 3) activity not reaching 90% of the histogram-based cumulative signal intensity was discarded. The correlation analysis with the fMRI connectivity values was performed taking sub-traces of 180 s. This is equivalent of 120 data points in the BOLD signal, above the minimum of 100 points that has been shown to be necessary in the analysis of dynamic functional connectivity between 0.01 and 0.2 Hz with an overlap of 66% (*Leonardi and Van De Ville, 2015*). To avoid spurious results in the correlation analysis given by the overlap of the collected data points, we performed a permutation test for each individual dataset, creating 10,000 data surrogates shuffling the connectivity indexes with the slow wave events in each subject. The normal distribution of the obtained r-scores represent the chance level of a spurious correlation and the mean of the distribution, the degree of intra-dependence. Thus, the obtained r-score is considered as significant when it falls into the 5% of the extremes of the distribution. The repeated measures correlation was performed using R software for statistical computing (*R Development Core Team, 2010*) using the rmcorr package (*Bakdash and Marusich, 2017*).

## Statistics

All statistical analysis were conducted using the statistical and machine learning toolbox of Matlab (The Mathworks, Inc, Natick, MA). Statistical measures were calculated in the parametric space, when attributes of data distributions, normality and homoscedasticity criteria allowed it. Statistical analysis comparing both conditions (slow wave and persistent activity) were conducted using paired t-test (same animals, different conditions) and plotted using the *boxplot* function over the data points distribution. Values are reported as mean ± SEM for normally distributed data.

## Data availability

All data generated or analyzed during this study are included in the manuscript and supporting files. All source data files are available on Dryad Digital repository (https://doi.org/10.5061/dryad. vmcvdncqk).

All custom Matlab codes used in these analyses are available at https://github.com/Strohlab/connectivityelife (*Aedo-Jury and Stroh, 2020*; copy archived at https://github.com/elifesciences-publications/connectivityelife).

## Acknowledgements

This work was supported by the DFG (SPP 1665: Resolving and manipulating neuronal networks in the mammalian brain - from correlative to causal analysis; CRC1193: Neurobiology of resilience to stress-related mental dysfunction: from understanding mechanisms to promoting preventions) and the Focus Program translational Neurosciences (FTN-Mainz). Authors thank Nuse Afahaene for excellent technical assistance and Eduardo Rosales Jubal for support in the Graph theory analyses.

## Additional information

### Funding

| Funder | Grant reference number | Author |
| --- | --- | --- |
| Deutsche Forschungsgemeinschaft | SFB 1193 | Albrecht Stroh |
| Deutsche Forschungsgemeinschaft | SPP 1665 | Albrecht Stroh |

The funders had no role in study design, data collection and interpretation, or the decision to submit the work for publication.

### Author contributions

Felipe Aedo-Jury, Conceptualization, Data curation, Writing; Miriam Schwalm, Conceptualization, Formal analysis, Writing; Lara Hamzehpour, Data curation, Software, Writing - review and editing; Albrecht Stroh, Conceptualization, Funding acquisition, Writing

### Author ORCIDs

Felipe Aedo-Jury (iD) https://orcid.org/0000-0002-8642-6955
Miriam Schwalm (iD) https://orcid.org/0000-0003-4162-2298
Albrecht Stroh (iD) https://orcid.org/0000-0001-9410-4086

### Ethics

Animal experimentation: Animal husbandry and experimental manipulation were carried out according to animal welfare guidelines of the Johannes Gutenberg-University Mainz and were approved by the Landesuntersuchungsamt Rheinland-Pfalz, Koblenz, Germany. (G14-1-040).

### Decision letter and Author response

Decision letter https://doi.org/10.7554/eLife.53186.sa1
Author response https://doi.org/10.7554/eLife.53186.sa2

## Additional files

### Supplementary files
• Transparent reporting form

### Data availability
All data generated or analysed during this study are included in the manuscript and supporting files. All source data files are available on Dryad Digital repository (https://doi.org/10.5061/dryad. vmcvdncqk). All custom Matlab codes used in these analyses are available at https://github.com/ Strohlab/connectivityelife (copy archived at https://github.com/elifesciences-publications/connectivi-tyelife; Aedo-Jury & Stroh, 2020).

The following dataset was generated:

| Author(s) | Year | Dataset title | Dataset URL | Database and Identifier |
|---|---|---|---|---|
| Aedo-Jury F, Schwalm M, Ham-zehpour L, Stroh A | 2020 | Brain states govern the spatio-temporal dynamics of resting-state functional connectivity | https://doi.org/10.5061/ dryad.vmcvdncqk | Dryad Digital Repository, 10.5061/ dryad.vmcvdncqk |

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
