## [Decision Letter]

Thank you for submitting your article "Brain states govern the spatio-temporal dynamics of resting state functional connectivity" for consideration by *eLife*. Your article has been reviewed by three peer reviewers, and the evaluation has been overseen by a Reviewing Editor and Christian Büchel as the Senior Editor. The reviewers have opted to remain anonymous.

The reviewers have discussed the reviews with one another and the Reviewing Editor has drafted this decision to help you prepare a revised submission.

Summary:

In this study, the authors reproduce and expand their previous work showing that functional connectivity in the rat brain differs between persistent state, as induced by medetomidine anesthesia, and slow wave state, as induced by deep isoflurane anesthesia. They expand the common ICA analysis to show that there are more significant correlations between brain regions under isoflurane than under medetomidine and that the networks under isoflurane resemble a random graph, while those under medetomidine resemble small world networks. Further, the authors provide evidence for their hypothesis that down-up transitions (as they call the onset of a slow wave event) of neuronal excitability as found in slow wave state drive functional connectivity. To probe this, they employ an unusual combination of calcium recordings and resting state fMRI. Combining these two approaches is challenging. In four of five animals they find a correlation between the number of significantly correlated brain regions and the number of down-up transitions. This result has important implications for resting state fMRI, since it calls for strict control of the brain state during resting state fMRI measurements.

Essential revisions:

As you can read from the following comments, there are two recurrent themes: (a) your text needs to be revised to make it easier to understand for the general reader. (b) you need to specify the statistics used to analyze your data.

1) Many of the observations reported in this study have previously been described (e.g. increasing cortical synchrony within high iso condition vs low iso or vs medetomidine (e.g. PMID: 20530220, Grandjean et al., 2014), network connectivity strength between conditions (Bukhari et al., 2018), graph theory metrics (D'Souza et al., 2014), frequency/power distribution (Paasonen et al., 2018, Grandjean et al., 2014), reduced connectivity when global signal is removed especially when there is global connectivity synchrony in high-iso (Grandjean et al., 2014)). Please cite these published findings and relate to the present study.

2) The combination of BOLD fMRI with the slow-rave recording by photometry (Figure 6B) is novel. However, there, a few methodological concerns that need to be addressed. Most notably, the concept of independence is violated in Figure 6B, because the sampled variable (number of slow-wave and number of significant edges) overlap in their sliding windows approach (3 min windows with 1 min overlap with both the previous and next window). Thus, in a statistical test, the samples may not be considered independent from one another, a key tenant of statistics.

3) The work would gain importance, if these results were reproduced in a larger group of animals. In particular, graph theory analysis is usually performed in larger cohorts of animals (or at least data sets). However, additional data analysis can also strengthen the paper substantially.

4) The text is often unclear, lacks proper definitions, there is wrong punctuation in several places making it difficult to read (some examples below). Specifically there are some terms that are not precise or incorrect in the text. For example, the Abstract refers to "these bimodal recordings". It is not clear to what this refers. Next the Abstract refers to a "persistent state". Normally one refers to "persistent activity", and maybe after a classification, it can be referred to as "the state with persistent activity" or even the persistent state.

4.1) Along these lines: Some sentences in the manuscript are rather confusing, or even wrong, for example in the Introduction there is: "The concept of brain states as defined by the local and global features of network dynamics by no means entails, that different behavioural states as e.g. anesthetized condition or natural sleep would be identical, however, defining features of the slow wave state, such as bimodality, can be identified in a plethora of different conditions as the fundamental difference between fast and slow timescale cortical dynamics is likely regulated by distinct mechanisms at the single neuron level". This needs rewriting.

4.2) With regards to revising the text: In the Introduction it is said: "Surprisingly, in rodents as in humans, little has been investigated on how the brain's current state impacts functional connectivity derived from BOLD activity, despite evidence that, for instance, small changes in anaesthesia can dramatically change brain connectivity ". This suggests that, if small changes in anaesthesia result in dramatic changes in brain connectivity, different brain states are -not surprisingly- going to have an impact on functional connectivity (as it has been shown, although questions remain open).

4.3) The Results start by referring to " We first conducted independent component (IC) analyses of rat task-free fMRI signals in the isoflurane-induced condition, and in the medetomidine-induced condition." Then one wonders, why in medetomidine? Lacking the information that medetomidine is the way to induce "persistent state" (given that the experimental paradigm in this regard was not described earlier). There is a poor definition of the "persistent state" in the presence of medetomidine. It is obviously a different brain state from slow waves, but what kind of brain state? How different is it from the awake? To what extent is it dose-dependent (as indeed it is)? If there is a detailed characterization by the authors or in the literature, it should be better reported, describing what is the state induced by medetomidine. It should be included at the start of the results including what the comparison is, and why medetomidine is a good equivalent of awake.

4.4) In the Discussion: "In contrast, animals in slow wave state showed fast, and temporally irregular transitions. This suggests, that individual, specific events within the slow wave state drive the state-specific network signature The occurrence of these events, down-up transitions, or slow waves, are not occurring in a regular, oscillation-like manner, but depend on the overall physiological state of the animal, such as the depth of anaesthesia.". The concept of "temporally irregular transitions" is confusing. The frequency of the slow oscillations may vary with depth of anaesthesia, however this is compatible with regularity in the oscillations.

4.5) In the Discussion: "Although, there is abundant evidence on the relevance of infra-slow electrical brain activity for resting state connectivity there has been no direct neural evidence so far linking brain states to changes in network configuration". Indeed, there are articles both for humans and for rodents illustrating different connectivity patterns with different brain states. However, the relation with the infra-slow (unmentioned so far) here is not that clear.

4.6) In the Discussion: "pathophysiological condition such as early AD", since the acronym AD was never defined earlier in the text, please use the explicit term.

4.7) Discussion: "In this study, we demonstrated that brain states can modify properties of the functional connectome through changes in cortical excitability." We do not know that it is just "excitability", since excitability is a broad term. There are other concomitant differences, such as dynamics.

4.8) The study of the cross-correlations during down to up states is highly suggestive but needs more robust and obvious evidence than that provided.

5) Description and presentation of the ICA is currently not sufficient.

5.1) More details on how ICA was performed are required: e.g. how many components were allowed; which components were assigned to noise, and how was this done; was breathing or heart rate used as regressor;

5.2) Presentation of ICA results is not clear: how many components were assigned to brain activity; what were the weights of these components; showing only three slices is not sufficient – include all slices relevant slices of the respective networks.

5.3) How was the DMN defined (ie which brain regions were included)? How many of these were found in the ICA component(s) and in which? Suggestion: perhaps register the ICA results on the brain atlas and provide quantitative data on how much of the respective network was identified – how were the average z-scores there.

5.4) To better illustrate the difference between the brain states, the authors may show serial slices of the brain for SW and P state, so the reader can see which regions were identified.

6) Since the activity of different networks in the different brain states is the fundamental point of this paper, the authors should provide a confirmation of their ICA, by performing a ROI based correlation analysis. This should be easily doable with the data and the atlas the authors have used already. Such independent confirmation of active networks in the different brain states would make a strong point and probably silence most of the doubt in the community.

7) Graph theory provides far more information than global results as shown in Figure 4. It would be extremely interesting to see how communities actually are different between brain states. Please provide a representation (e.g. a 2D plot using a force-based algorithm) of the networks in the two brain states.

8) There is a dramatic difference in noise level in the calcium traces in Figure 5 between SW-I and P-M. Is the scale bar with 3% valid for all three traces or only for the top one? Are these data from the same animal, were they acquired in the same session? If yes, the anesthesia switch needs to be described in both Materials and methods and caption. Add scale bars for each trace. Explain experimental details in the caption. Since there seems to be substantial variation in noise levels, the authors should provide exemplary calcium traces for all conditions for all animals (as a Supplement).

9) Why was 0.5-0-7% isoflurane added to medetomidine? This appears arbitrary. Is anything known about which concentration induces brain state changes. How does isoflurane concentration influence networks or hemodynamic and calcium signal. Please discuss.

10) Results shown in Figure 6Bii should be additionally validated by performing correlation plots not only with number of events but also with 1) total up-time during the 180s, 2) average amplitude during the 180s, 3) position of the 180s-frame to exclude an effect of time during 30 min. Please show data also for the fifth animal.

11) What is the meaning of cross correlation amplitude of filtered data as shown in Figure 7D and E?

12) Figure captions do not provide sufficient information. Make sure that all symbols are explained, and experimental conditions become clear – for all figures. Three examples: Explain what is shown in the boxplots. What do lines between left and right data in box plots indicate? What do shaded areas indicate (SD, SEM,…)?

Related to this, there is no statistics described in the Materials and methods section. Adding details is very important to determine what test and how it was performed. Without such a section, the reader cannot assess the plausibility of the results, and thus cannot provide in-depth comments pertaining to the results.

13) Connectivity matrices must be labelled to identify the individual brain regions. Add a legend on the matrix or a color code plus legend (probably best as a supplement).

14) Were both positive and negative correlations used. For graph theory analysis is usually separated for positive and negative correlations.

15) Please explain in detail the process of adjacency matrices construction and the calculation of the threshold.

16) Connectivity analysis, second paragraph: Please specify the procedure of FDR. Please give the correlation value of connectivity strength and distance not only its significance.

17) There is no information about the topography of the FC. It is shown that the correlation between Euclidean distance is more negative in slow wave state. It could illustrate that during slow wave state the activity is propagating so the FC follows a spatial gradient rather than long-range connections. Actually, in Figure 2A, one can see that the interhemispheric connections are weaker in slow-wave state than in persistent state, although the overall correlation magnitude is larger.

18) In Figure 2A, 5C, 6A and 6B, please consider labelling the axis of the matrices, not with all the 96 ROIs but at least partially to give the reader a better idea of the average connectivity found in the two states.

19) Figure 2C: please describe the test used both in text and figure caption.

Figure 3: missing the number of the individuals in the figure legend.

20) In the section Connectivity differences are reflecting different brain states: "In this case, the resulting values of persistent state induced by low isoflurane…. (Figure 4D) (t(5)=5.19, p=0.003). We used the same matrix similarity computation in a dynamic connectivity analysis […] than in the case of slow wave state induced by high isoflurane concentrations (Figure 4E). ". There is an incorrect figure reference, it must be Figure 5D and 5E.

21) In Figure 7B and C, are those recordings from V1 or S1? Consider showing both together for consistency and to show the propagation delay. Maybe different colours can be used for V1 and S1 and different shades for the filter's frequency bands.

---

## [Author Response]

Essential revisions:As you can read from the following comments, there are two recurrent themes: (a) your text needs to be revised to make it easier to understand for the general reader. (b) you need to specify the statistics used to analyze your data.

We appreciate the careful and dedicated revision of our work. We indeed agree with all suggestions made to improve this manuscript and have proceeded accordingly. We addressed all comments, please see below. These are the two main revisions:

1) We have included new experiments, increasing the number of animals, strengthening our statistics. In addition, we augmented the statistical methods used for analyses.

2) We substantially improved the description of the methodologies used in this study.

1) Many of the observations reported in this study have previously been described (e.g. increasing cortical synchrony within high iso condition vs low iso or vs medetomidine (e.g. PMID: 20530220, Grandjean et al., 2014), network connectivity strength between conditions (Bukhari et al., 2018), graph theory metrics (D'Souza et al., 2014), frequency/power distribution (Paasonen et al., 2018, Grandjean et al., 2014), reduced connectivity when global signal is removed especially when there is global connectivity synchrony in high-iso (Grandjean et al., 2014)). Please cite these published findings and relate to the present study.

We thank the reviewer for this comment and have revised the text accordingly. We included the mentioned literature and relate our results to these important previous findings.

2) The combination of BOLD fMRI with the slow-rave recording by photometry (Figure 6B) is novel. However, there, a few methodological concerns that need to be addressed. Most notably, the concept of independence is violated in Figure 6B, because the sampled variable (number of slow-wave and number of significant edges) overlap in their sliding windows approach (3 min windows with 1 min overlap with both the previous and next window). Thus, in a statistical test, the samples may not be considered independent from one another, a key tenant of statistics.

We agree with the reviewer on the point of sample independence and thus, have conducted two new analyses for this data. First, we performed a permutation creating 10000 surrogates, shuffling the data pairs of each subject. The means of the correlations scores (r) thereby reflect the degree of dependency of the data, i.e. the intrinsic bias. We compared the obtained r-score with the distribution of the permutation test and found significant results for 5 out of 6 animals. Furthermore, we conducted a repeated measures correlation analysis as proposed by Bakdash and Marusich, 2017, which also yielded significant results.

3) The work would gain importance, if these results were reproduced in a larger group of animals. In particular, graph theory analysis is usually performed in larger cohorts of animals (or at least data sets). However, additional data analysis can also strengthen the paper substantially.

We have proceeded with the recommendation of the reviewer and have included a new data set, increasing the number of animals from 8 to 15 for the fMRI connectivity analysis (including graph theory analysis) and from 5 to 6 for the combined optic fiber recordings and fMRI measurements.

4) The text is often unclear, lacks proper definitions, there is wrong punctuation in several places making it difficult to read (some examples below). Specifically there are some terms that are not precise or incorrect in the text. For example, the Abstract refers to "these bimodal recordings". It is not clear to what this refers. Next the Abstract refers to a "persistent state". Normally one refers to "persistent activity", and maybe after a classification, it can be referred to as "the state with persistent activity" or even the persistent state.

We have thoroughly revised the text according to this comment. We are using “persistent activity” now in agreement with more common terminology and have rewritten or clarified the mentioned parts.

4.1) Along these lines: Some sentences in the manuscript are rather confusing, or even wrong, for example in the Introduction there is: "The concept of brain states as defined by the local and global features of network dynamics by no means entails, that different behavioural states as e.g. anesthetized condition or natural sleep would be identical, however, defining features of the slow wave state, such as bimodality, can be identified in a plethora of different conditions as the fundamental difference between fast and slow timescale cortical dynamics is likely regulated by distinct mechanisms at the single neuron level". This needs rewriting.

This section of the Introduction has been replaced entirely.

4.2) With regards to revising the text: In the Introduction it is said: "Surprisingly, in rodents as in humans, little has been investigated on how the brain's current state impacts functional connectivity derived from BOLD activity, despite evidence that, for instance, small changes in anaesthesia can dramatically change brain connectivity ". This suggests that, if small changes in anaesthesia result in dramatic changes in brain connectivity, different brain states are -not surprisingly- going to have an impact on functional connectivity (as it has been shown, although questions remain open).

We revised the text in the Introduction and now wrote:

“…Despite evidence of subtle changes in anesthesia level being able to dramatically change brain connectivity^43-45^, little has been investigated on how the brain’s current state impacts functional connectivity derived from BOLD activity. While variations in functional connectivity have been related to cognitive state in humans^46^, the effect of neurophysiologically defined states on functional connectivity in the cortex remains unknown.”

4.3) The Results start by referring to " We first conducted independent component (IC) analyses of rat task-free fMRI signals in the isoflurane-induced condition, and in the medetomidine-induced condition." Then one wonders, why in medetomidine? Lacking the information that medetomidine is the way to induce "persistent state" (given that the experimental paradigm in this regard was not described earlier). There is a poor definition of the "persistent state" in the presence of medetomidine. It is obviously a different brain state from slow waves, but what kind of brain state? How different is it from the awake? To what extent is it dose-dependent (as indeed it is)? If there is a detailed characterization by the authors or in the literature, it should be better reported, describing what is the state induced by medetomidine. It should be included at the start of the results including what the comparison is, and why medetomidine is a good equivalent of awake.

We now introduce the concept of persistent activity as a state which can occur during light sedation (with e.g. medetomidine) already in the Introduction (“…In a state of persistent activity, which can occur during periods of light anesthesia, sedation or during awake periods^13^, neurons are rather depolarized, sparsely active, leading to temporally dynamic, modality specific, network configurations^19,20^”) and added a sentence in the results stating the components we identified under medetomidine-induced persistent activity and in which conditions these have been described before by others (“…the identified components for persistent activity resembled those described by others for awake^49,62^ or sedated rodents^1,63^.”).

4.4) In the Discussion: "In contrast, animals in slow wave state showed fast, and temporally irregular transitions. This suggests, that individual, specific events within the slow wave state drive the state-specific network signature The occurrence of these events, down-up transitions, or slow waves, are not occurring in a regular, oscillation-like manner, but depend on the overall physiological state of the animal, such as the depth of anaesthesia.". The concept of "temporally irregular transitions" is confusing. The frequency of the slow oscillations may vary with depth of anaesthesia, however this is compatible with regularity in the oscillations.

We agree with the reviewer that this section was confusing and have rewritten this sentence in the Discussion incorporating the reviewer’s suggestions regarding the regularity being dependent on the depth of anesthesia: “…This suggests that individual specific events within slow wave activity could drive state-specific network signatures, while it has to be taken into account that the regularity of down-up transitions or slow wave events can depend on the overall physiological condition of the animal, such as the depth of anesthesia^6^.”

4.5) In the Discussion: "Although, there is abundant evidence on the relevance of infra-slow electrical brain activity for resting state connectivity there has been no direct neural evidence so far linking brain states to changes in network configuration". Indeed, there are articles both for humans and for rodents illustrating different connectivity patterns with different brain states. However, the relation with the infra-slow (unmentioned so far) here is not that clear.

We now further clarify our intended statement in this sentence: “…Although, there is abundant evidence on the relevance of, for example, infra-slow electrical brain activity for resting state connectivity^37,76,77^, there so far has been no evidence linking different population activity states of the brain to changes in functional network configuration directly.” Although for example infra-slow activity (terminology of the cited studies) has been previously studied in relation to resting state activity, we are unaware of studies directly relating activity states of local neural populations to changes in functional network configurations of the entire brain.

4.6) In the Discussion: "pathophysiological condition such as early AD", since the acronym AD was never defined earlier in the text, please use the explicit term.

AD has been replaced with Alzheimer’s disease in the revised version of the manuscript.

4.7) Discussion: "In this study, we demonstrated that brain states can modify properties of the functional connectome through changes in cortical excitability." We do not know that it is just "excitability", since excitability is a broad term. There are other concomitant differences, such as dynamics.

We agree with the reviewer and have removed the term excitability from this sentence in the Discussion.

4.8) The study of the cross-correlations during down to up states is highly suggestive but needs more robust and obvious evidence than that provided.

In this study, we relate a well-defined neurophysiological event, a slow-oscillation-associated slow wave, to the hemodynamic signal, and contrast the brain state governed by these slow waves with a brain state characterized by rather persistent activity. For that we used, as pioneered in the 2017 *eLife* manuscript (Schwalm et al.), a regressor-based approach. Here, we collected important new evidence on the impact of the temporal dynamics of ongoing activity to the hemodynamic response. But, this approach has limitations: we relate two physiological signals with drastically different temporal resolutions: 1 ms in terms of the optical recordings, 1 s in terms of the fMRI recordings; and we are dealing with different origins of the signals: action potential-mediated influx of calcium vs. a hemodynamic response. We strongly believe, that we provided meaningful further evidence of the relation of these two signals, and that the field of small animal “resting state”, task-free fMRI needs to move beyond comparing anesthetic regimens. But, these inherent limitations prevail, and we simply do not see, how we could provide “obvious evidence” in this complex relation of two drastically different physiological signals.

5) Description and presentation of the ICA is currently not sufficient.

We now include a significantly improved analysis of the independent components, as well as a more detailed description thereof, please check the following points.

5.1) More details on how ICA was performed are required: e.g. how many components were allowed; which components were assigned to noise, and how was this done; was breathing or heart rate used as regressor;

We have added this information in the Materials and methods section.

5.2) Presentation of ICA results is not clear: how many components were assigned to brain activity; what were the weights of these components; showing only three slices is not sufficient – include all slices relevant slices of the respective networks.

We agree with the reviewer that a more detailed presentation of the ICA was needed. For this reason, we have added a new supplementary figure with the maps of the selected component for each animal. We have also contrasted the found activation maps with atlas-based information. We also extracted the ROIs of the selected map and performed a connectivity analysis to double check the veracity of the obtained networks.

5.3) How was the DMN defined (ie which brain regions were included)? How many of these were found in the ICA component(s) and in which? Suggestion: perhaps register the ICA results on the brain atlas and provide quantitative data on how much of the respective network was identified – how were the average z-scores there.

We have taken the suggestion of the reviewer into account and added the information on the exact DMN regions we found. In a new supplementary figure, we have now also added the maps of each animal that reflect the DMN. This now gives the reader a clear idea on the revealed components of the ICA and their localization.

5.4) To better illustrate the difference between the brain states, the authors may show serial slices of the brain for SW and P state, so the reader can see which regions were identified.

In Figure 1, we show renderings in all three anatomical orientations, in which, at least in our view, the different activity patterns can be clearly delineated. We would not see how to display our findings in a way which would add further information.

6) Since the activity of different networks in the different brain states is the fundamental point of this paper, the authors should provide a confirmation of their ICA, by performing a ROI based correlation analysis. This should be easily doable with the data and the atlas the authors have used already. Such independent confirmation of active networks in the different brain states would make a strong point and probably silence most of the doubt in the community.

We agree with the reviewer on this point and have conducted the suggested analysis in the new version of Figure 1.

7) Graph theory provides far more information than global results as shown in Figure 4. It would be extremely interesting to see how communities actually are different between brain states. Please provide a representation (e.g. a 2D plot using a force-based algorithm) of the networks in the two brain states.

We have included a new subplot in Figure 3 that contains all the pairs of significant connections over the cortex in both states. We believe this is even more informative in terms of visualization than a force-based algorithm in terms of the patterns of connectivity in each state.

8) There is a dramatic difference in noise level in the calcium traces in Figure 5 between SW-I and P-M. Is the scale bar with 3% valid for all three traces or only for the top one? Are these data from the same animal, were they acquired in the same session? If yes, the anesthesia switch needs to be described in both methods and caption. Add scale bars for each trace. Explain experimental details in the caption. Since there seems to be substantial variation in noise levels, the authors should provide exemplary calcium traces for all conditions for all animals (as a Supplement).

Since the data collection for the different brain states was performed in different sessions, the data previously presented does not correspond to the same experiment. The signal-to-noise-level varies between experiments, as it is dependent on 1) the staining efficacy, which is different from each injection pipette used, and, the relative placement of the optical fiber to the stained region. This is inherent to method, please see also previous studies of us and others (Stroh et al., 2013; Busche,…Konnerth, Science 2008). Therefore, normalization approaches need to be implemented, and amplitudes cannot be compared between experiments. Nevertheless, we now included a trace of an experiment for which the anesthesia was modified within the same session. Thus, this new dataset represents exactly the same noise levels for the calcium recordings.

9) Why was 0.5-0-7% isoflurane added to medetomidine? This appears arbitrary. Is anything known about which concentration induces brain state changes. How does isoflurane concentration influence networks or hemodynamic and calcium signal. Please discuss.

We apologize that this part was not clear in the text. The condition called “low isoflurane” included *only* 0.5-0.7% of isoflurane and no added medetomidine. This allowed to keep the animal in stable persistent activity in the scanner while using the same anesthetic that can induce the slow wave activity. We have rephrased this part accordingly to make it cleared for the reader.

10) Results shown in Figure 6Bii should be additionally validated by performing correlation plots not only with number of events but also with 1) total up-time during the 180s, 2) average amplitude during the 180s, 3) position of the 180s-frame to exclude an effect of time during 30 min. Please show data also for the fifth animal.

We have included one more animal in our analysis and are now plotting data for 6 animals. Data on the total up-time and the average amplitude is highly dependent on the staining quality and the fiber placement, please see above. Due to bleaching effects OGB-1 signal quality can deteriorate over time, leading to smaller signal to noise ratio that influences the length and amplitude of the slow waves. Finally, taking into consideration that the TR of the fMRI is 1.5 seconds, this seems to be irrelevant for the BOLD signal differences as it only affects a few milliseconds of slow wave duration.

11) What is the meaning of cross correlation amplitude of filtered data as shown in Figure 7D and E?

The purpose of this analysis is to show that down-up transitions drive the high correlation between cortical areas by using a signal of neural origin. We have rephrased this part of the results in order to make it more accessible to the reader.

12) Figure captions do not provide sufficient information. Make sure that all symbols are explained, and experimental conditions become clear – for all figures. Three examples: Explain what is shown in the boxplots. What do lines between left and right data in box plots indicate? What do shaded areas indicate (SD, SEM,…)?

We have now added a more detailed description of each figure in the figure legends.

Related to this, there is no statistics described in the method section. Adding details is very important to determine what test and how it was performed. Without such a section, the reader cannot assess the plausibility of the results, and thus cannot provide in-depth comments pertaining to the results.

We now explain the statistical tests used in the text as well as in the figure captions. Accordingly, we have also added the statistical methods in the Materials and methods section of the manuscript.

13) Connectivity matrices must be labelled to identify the individual brain regions. Add a legend on the matrix or a color code plus legend (probably best as a supplement).

We have added a supplementary figure containing the name of all regions used in the study as well as their position in the matrices.

14) Were both positive and negative correlations used. For graph theory analysis is usually separated for positive and negative correlations.

Following the indications of Rubinov and Sporns, 2010 for the construction of adjacency matrices when analyzing fMRI data, we used only the positive correlation values. We have added this information now in the Materials and methods section.

15) Please explain in detail the process of adjacency matrices construction and the calculation of the threshold.

We have added detailed information in the Materials and methods section, explaining the threshold calculation using the FDR calculated for each individual, and the Weighted Undirected Networks construction using Sporns Matlab toolbox for graph theory analysis.

16) Connectivity analysis, second paragraph: Please specify the procedure of FDR. Please give the correlation value of connectivity strength and distance not only its significance.

We have added a sentence in the connectivity analysis section explaining how the FDR was calculated and used in our analyses.

17) There is no information about the topography of the FC. It is shown that the correlation between Euclidean distance is more negative in slow wave state. It could illustrate that during slow wave state the activity is propagating so the FC follows a spatial gradient rather than long-range connections. Actually, in Figure 2A, one can see that the interhemispheric connections are weaker in slow-wave state than in persistent state, although the overall correlation magnitude is larger.

We have added a subplot in Figure 3 that contains all the pairs of significant connections for the cortex in both states. We believe this is highly informative in terms of visualization. We have also added a supplementary figure showing the distribution and names of the regions and their position in the matrices.

18) In Figure 2A, 5C, 6A and 6B, please consider labelling the axis of the matrices, not with all the 96 ROIs but at least partially to give the reader a better idea of the average connectivity found in the two states.

We have added a supplementary figure containing all the labels.

19) Figure 2C: please describe the test used both in text and figure caption.

We have added the statistical tests used in both the figure legend and the main text.

Figure 3: missing the number of the individuals in the figure legend.

We have added the number.

20) In the section Connectivity differences are reflecting different brain states: "In this case, the resulting values of persistent state induced by low isoflurane…. (Figure 4D) (t(5)=5.19, p=0.003). We used the same matrix similarity computation in a dynamic connectivity analysis […] than in the case of slow wave state induced by high isoflurane concentrations (Figure 4E). ". There is an incorrect figure reference, it must be Figure 5D and 5E.

This has been changed in the new version of the manuscript.

21) In Figure 7B and C, are those recordings from V1 or S1? Consider showing both together for consistency and to show the propagation delay. Maybe different colours can be used for V1 and S1 and different shades for the filter's frequency bands.

We have added the region (V1) in the figure legend and we have improved our explanation in the body of the text. We hope this helps to clarify.